# Numerical evaluation of two-time correlation functions in open quantum systems with matrix product state methods: a comparison

Stefan Wolff[1], Ameneh Sheikhan[1], Corinna Kollath[1*]

**1** Physikalisches Institut, University of Bonn, Nussallee 12, 53115 Bonn, Germany
*corinna.kollath@uni-bonn.de

September 17, 2020

## Abstract

We compare the efficiency of different matrix product state (MPS) based methods for the calculation of two-time correlation functions in open quantum systems. The methods are the purification approach [1] and two approaches [2, 3] based on the Monte-Carlo wave function (MCWF) sampling of stochastic quantum trajectories using MPS techniques. We consider a XXZ spin chain either exposed to dephasing noise or to a dissipative local spin flip. We find that the preference for one of the approaches in terms of numerical efficiency depends strongly on the specific form of dissipation.

# 1  Introduction

The investigation of open quantum many-body systems has been a very active field of research over the past decades. One motivation is the understanding of destructive effects of environments on quantum processes which are used in quantum technologies, e.g in quantum computing or quantum communication. More recently, another point of view has been taken. Environments are particularly tailored in order to stabilize and control quantum many-body states [4–8].

However, it has been shown that the dynamic properties of such states can have very distinct behavior from their Hamiltonian counterparts [9]. In particular the response of open quantum systems to external perturbations very different to the Hamiltonian evolution.

Quantities which are of particular importance in this respect are two-time correlation functions $\langle B(t_2)A(t_1)\rangle$. Here $A$ and $B$ are operators, $t_1$ and $t_2$ are two different times, and $\langle\ldots\rangle = \mathrm{tr}(\rho\ldots)$ is the expectation value over the density matrix $\rho$ of a given system. In isolated systems, such two-time correlation functions are powerful tools to give information on the response of the system to a small perturbation. Many experimental techniques are based on such processes and the observation of the subsequent response is described by these two-time functions. Examples include neutron scattering [10], ARPES [11], conductivity and magnetization measurements in solids [12] and spectroscopic measurement as radio-frequency [13], Raman, Bragg [14] or modulation spectroscopy [15] in the field of quantum gases.

Two-time correlations have been studied extensively in isolated many-body quantum systems both in and out of equilibrium. However, in many-body quantum systems coupled to environments their determination is very challenging and only few studies are available mostly using approximate approaches or small systems [9, 16–20]. In particular, a change of the behavior of dynamic correlations has been demonstrated in the presence of dissipation (see for example [21–26]).

In this work we present a comprehensive study on the application of matrix product state (MPS) algorithms to the determination of two-time correlation functions in open systems. We compare an extension of the purification approach [27–34] to two-time correlations and two different stochastic approaches based on the unraveling of the quantum evolution [2, 3, 36–39, 70]. The first approach has been proposed by Breuer et al. [2] and the second approach by

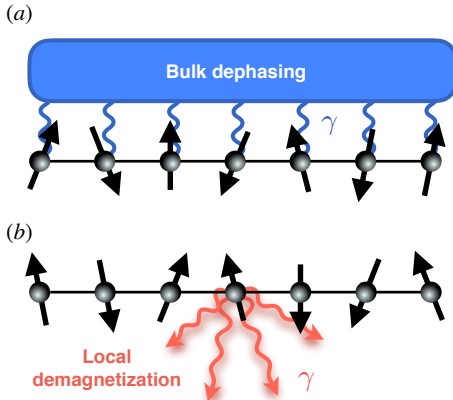

Figure 1: The sketch of the spin models with (*a*) bulk dephasing where all spins are coupled to the bath and (*b*) local demagnetization where the central spin is disspative.

Mølmer et al. [3]. The comparison is performed using an XXZ spin model with two different couplings to the environment. The first coupling is a dephasing noise applied globally to the system and the second a local loss of magnetization. In section 2 we describe the models used. In section 3 we give an introduction to matrix product state based methods for open quantum systems. In section 4, we introduce the three different methods in order to calculate two-time correlation functions and in section 5 we present a comprehensive comparison between the methods.

## 2 Model

We consider spin-1/2 chains, with coupling between spins on adjacent sites, as a paradigmatic model for interacting one-dimensional systems. In addition, the system is subjected to an environment which introduces dissipative processes to the system dynamics. Assuming that retroactive influences of earlier dissipation effects on the current dynamics can be neglected, the system dynamics can be described by the Lindblad master equation [39]

$$\frac{\partial}{\partial t}\rho(t) = \mathcal{L}\rho(t) = -\frac{i}{\hbar}\left[H_{\text{XXZ}}, \rho(t)\right] + \mathcal{D}[\rho(t)] \tag{1}$$

with the superoperator $\mathcal{L}$ and the system density matrix $\rho$. The first term on the right hand side represents the unitary contribution generated by the Hamiltonian. Here we consider the XXZ spin-1/2 Hamiltonian

$$H_{\text{XXZ}} = \sum_{l=1}^{L-1}\left[J_x\left(S_l^x S_{l+1}^x + S_l^y S_{l+1}^y\right) + J_z S_l^z S_{l+1}^z\right], \tag{2}$$

describing a chain of $L$ spins, where $J_x$ and $J_z$ are exchange couplings according to different spin directions and $S_l^\alpha$ is the spin operator in direction $\alpha$ at site $l$. In equilibrium this model is well understood and exhibits in the ground state, three phases for different ratios of the interaction strengths [40, 41]: For $-1 \leq J_z/J_x \leq 1$ a gapless Tomonaga-Luttinger liquid is formed, whereas $J_z/J_x < -1$ and $J_z/J_x > 1$ present gapped phases showing ferromagnetic

and antiferromagnetic nature, respectively. The second term on the right hand side of Eq. 1 represents dissipative noise. We use the Lindblad form of the dissipator which is given by

$$\mathcal{D}[\rho] = \sum_l \Gamma_l \left( L_l \rho L_l^\dagger - \frac{1}{2} L_l^\dagger L_l \rho - \frac{1}{2} \rho L_l^\dagger L_l \right). \tag{3}$$

Here $\Gamma_l$ is the effective dissipation strength corresponding to the Lindblad jump operator $L_l$. For the comparison of methods presented here, two different types of dissipation are considered. As will be shown, each of them have different impacts on various aspects of the performance of the methods.

First we study systems exposed to bulk dephasing (see Fig. 1($a$)), where the jump operators $L_l$ are given by the set of local $S_l^z$ operators controlled by the dissipation strength $\Gamma_l = \gamma$

$$\mathcal{D}_1(\rho) = \gamma \sum_{l=1}^{L} \left( S_l^z \rho S_l^z - \frac{1}{4} \rho \right). \tag{4}$$

Here the jump operators $S_l^z$ are Hermitian, which implies that the infinite temperature state is a steady state of the model. One can show that this is the unique steady state. The dissipative dynamics arising in this and related models has been studied previously and interesting critical dynamics and aging dynamics have been pointed out to occur [1, 18, 42–44].

Furthermore, we investigate the case in which the coupling to the environment results in a local defect that is represented by the single Lindblad operator $S_c^-$ which only acts on the central site as shown in Fig. 1($b$). Here $c$ is the index of the central site for a chain with an odd number of sites and the center-left site otherwise. The dissipation strength is $\Gamma_c = \gamma$ and $\Gamma_{l \neq c} = 0$ which results in the dissipator

$$\mathcal{D}_2(\rho) = \gamma \left( S_c^- \rho S_c^+ - \frac{1}{2} S_c^+ S_c^- \rho - \frac{1}{2} \rho S_c^+ S_c^- \right). \tag{5}$$

The steady state of this system is the ferromagnetic state with all spins pointing down. Using the mapping to interacting fermions or hard core bosons, the jump operator corresponds to a local particle loss process. Similar 'lossy' defects have been studied in a variety of models previously and interesting transport effects and meta-stable states have been identified [45–55].

## 3   Matrix product state approaches for open quantum system dynamics

A variety of tensor network based algorithms has been successfully used to simulate the dynamics of open one-dimensional quantum systems with a focus on equilibrium or equal time properties. This section is structured such that we start by giving a short overview of the established concepts of MPS in closed quantum systems in Sec. 3.1, which are essential for the extension to open systems. Subsequently, we describe two prominent approaches for computing the dissipative dynamics of open quantum systems. We describe first the full evolution of the purified density matrix in Sec. 3.2 and second the Monte-Carlo wave function (MCWF) sampling of stochastic quantum trajectories in Sec. 3.3.

## 3.1 Matrix product state formalism for closed systems

The description of quantum states in MPS form has become a standard method for the simulation of one-dimensional many-body quantum systems. It has been used for a wide range of models, since it is very efficient and well-controlled approximation [56, 57]. In this section and the following on open systems, we describe the basics and key concepts of this technique [56] such that in the following we can detail the particularities of the approach to the determination of two-time functions in open quantum systems.

### 3.1.1 MPS representation of quantum many-body states

The idea relies on the approximate representation of the quantum many-body wave function for a one-dimensional lattice system of $L$ sites as a set of local tensors/matrices. In order to achieve this, a singular value decomposition (SVD) is applied to the amplitude matrix of a bipartite system whose subparts $A$ and $B$ are connected by a bond between site $l$ and $l+1$

$$|\psi\rangle = \sum_{m,n} \psi_{m,n} |m\rangle_A \otimes |n\rangle_B \overset{SVD}{=} \sum_{m,n,a_l} U_{ma_l} s_{a_l} V^\dagger_{a_l n} |m\rangle_A \otimes |n\rangle_B \,. \tag{6}$$

Here $|m\rangle_{A/B}$ are basis states in subsystem $A/B$ and $U$, $s$, $V$ are obtained by the singular value decomposition of the amplitude matrix $\psi_{m,n}$. Iterating this procedure on all bonds then yields the expression of the quantum state by single site tensors

$$|\psi\rangle = \sum_{\sigma_1,\ldots,\sigma_L} \sum_{a_1,\ldots,a_{L-1}} M^{\sigma_1}_{1,a_1} M^{\sigma_2}_{a_1,a_2} \ldots M^{\sigma_L}_{a_{L-1},1} |\vec{\sigma}\rangle \,, \tag{7}$$

where we use $|\vec{\sigma}\rangle = |\sigma_1 \sigma_2 \ldots \sigma_L\rangle$ and $\sigma_l$ labels the local basis states at the site $l \in \{1, 2, \ldots, L\}$. The number of local basis states is called the physical dimension $d$, where for the considered spin-1/2 model $d = 2$ and $\sigma_l \in \{\uparrow, \downarrow\}$. The representation 7 is still exact. The singular values $s_{a_l}$ are the coefficients of the Schmidt decomposition and are thus directly linked to the von Neumann entropy

$$S_{\mathrm{vN}} = -\sum_{a_l} s^2_{a_l} \log\left(s^2_{a_l}\right) \,. \tag{8}$$

The von Neumann entropy is a measure for the entanglement between the two subsystems connected by the considered bond. If the entanglement is not too strong, this corresponds to a sufficiently fast decay of the descendingly sorted squared singular values $s^2_{a_l}$. In this case, the dimension of the matrices for the representation of the state can be cut at a maximal value $D$, resulting in a compressed state which is a very good approximation of the exact state. A weak von Neumann entanglement is for example found for ground states of short-ranged one-dimensional Hamiltonians [56]. As the value $D$ limits the extend of the indices $\{a_1, a_2, \ldots, a_{L-1}\}$, connecting two adjacent site tensors, this parameter is known as the bond dimension. The approximation is controlled by the sum of the discarded squared singular values, the so-called truncation weight

$$\varepsilon = \sum_{a_l > D} s^2_{a_l}, \tag{9}$$

and reduces the computational complexity from exponential to polynomial.

In the following we will use the established graphical notation for tensor networks [56]

$$T_{ijk} \equiv \begin{array}{c} i \\ \bigcirc \\ j \quad k \end{array}, \tag{10}$$

where a tensor is portrayed by a shape (here circle) with sticking out lines representing the indices. Connecting two lines depicts a tensor contraction with regard to this pair of indices. Using this notation, a MPS is depicted by

$$|\psi\rangle = \sum_{\vec{\sigma}} \begin{array}{c} \sigma_1 \quad \sigma_2 \quad \sigma_3 \quad \cdots \quad \sigma_L \\ \bigcirc \overset{a_1}{-} \bigcirc \overset{a_2}{-} \bigcirc \overset{a_3 \ a_{L-1}}{\cdots\cdots} \bigcirc \end{array} |\vec{\sigma}\rangle. \tag{11}$$

### 3.1.2 Time-dependent matrix product state algorithm

Time-dependent matrix product state ($t$MPS) algorithms are well-established tools for the efficient computation of the dynamics of closed many-body quantum systems at zero [56–60] and finite temperature [61–64]. In particular, it is possible to calculate the time evolution of a MPS for the considered spin-1/2 model with $\gamma = 0$, using a Suzuki-Trotter decomposition [65, 66] of the time evolution operator for small time steps $\Delta t$. Here we use the second order Suzuki-Trotter decomposition given by

$$\mathrm{e}^{-iH\Delta t/\hbar} = \mathrm{e}^{-iH_{\mathrm{odd}}\Delta t/(2\hbar)}\mathrm{e}^{-iH_{\mathrm{even}}\Delta t/\hbar}\mathrm{e}^{-iH_{\mathrm{odd}}\Delta t/(2\hbar)} + \mathcal{O}(\Delta t^3). \tag{12}$$

Here $H_{\mathrm{odd}}$ ($H_{\mathrm{even}}$) only contains parts of the Hamiltonian covering the odd (even) numbered bonds of the lattice, so that all contributing terms are commuting, but $[H_{\mathrm{odd}}, H_{\mathrm{even}}] \neq 0$. In the diagram formalism, this corresponds to a successive application of two-site gate operations and compressions

$$|\psi(t + \Delta t)\rangle = \sum_{\vec{\sigma}} \begin{array}{c} \sigma_1 \ \sigma_2 \cdots \qquad\qquad \sigma_L \\ \hline \end{array} |\vec{\sigma}\rangle \quad + \mathcal{O}(\Delta t^3), \tag{13}$$

where the application order is indicated by the dotted arrow and the different colors distinguish gates and subsequent compressions of $H_{\mathrm{odd}}$ and $H_{\mathrm{even}}$. By contracting these gates with the MPS tensors, the bond dimension also increases from $D$ to $d^2 D$, so that a subsequent compression via SVD is necessary.

A large reduction of the computational effort can be achieved by including conservation laws. In the present case, the total magnetization $M_{\mathrm{tot}} = \sum_j S_j^z$ is preserved by the closed system evolution under the Hamiltonian $H_{\mathrm{XXZ}}$. This leads to a block-diagonal form of matrices, and thus, tensor operations, such as SVDs or tensor contractions, can be performed much more efficiently.

## 3.2 Purification approach for the evolution of the density matrix of an open quantum system

### 3.2.1 Purification of the density matrix

One way to transfer the techniques from the previous section to dissipative systems with finite dissipation strength $\gamma$, is to rewrite the density matrix acting on the physical Hilbert space $\mathcal{H}_{\text{phys}}$ as a state in a doubled space $\mathcal{H}_{\text{phys}} \otimes \mathcal{H}_{\text{phys}}$ [27–33, 56]

$$\rho = \sum_{\vec{\sigma}, \vec{\sigma}'} \rho_{\vec{\sigma}, \vec{\sigma}'} |\sigma_1 \sigma_2 \dots \sigma_L\rangle \langle \sigma_1' \sigma_2' \dots \sigma_L'|$$

$$\longrightarrow \quad |\rho\rangle\!\rangle = \sum_{\vec{\sigma}, \vec{\sigma}'} \rho_{\vec{\sigma}, \vec{\sigma}'} |\sigma_1 \sigma_1' \sigma_2 \sigma_2' \dots \sigma_L \sigma_L'\rangle\!\rangle \tag{14}$$

where $|\rho\rangle\!\rangle \in \mathcal{H}_{\text{phys}} \otimes \mathcal{H}_{\text{phys}}$. Thus, the density matrix acting on the physical space becomes a pure state in the 'doubled' space, giving the procedure the name purification. The order of the resorting is in principle arbitrary. We use here the given resorting of the state, since this has the advantage that the time-evolution only acts on four 'neighboring sites'. In this representation of the density matrix in the super space $\mathcal{H}_{\text{phys}} \otimes \mathcal{H}_{\text{phys}}$, the Lindblad master equation (Eq. 1) is written as,

$$\frac{\partial}{\partial t} |\rho(t)\rangle\!\rangle = \mathbb{L} |\rho(t)\rangle\!\rangle \equiv \left( -\frac{i}{\hbar} H \otimes I + \frac{i}{\hbar} I \otimes H^T + \mathbb{D} \right) |\rho(t)\rangle\!\rangle \tag{15}$$

with the new representation of the dissipator $\mathcal{D}$ in the super space,

$$\mathbb{D} \equiv \sum_l \Gamma_l \left( L_l \otimes (L_l^\dagger)^T - \frac{1}{2} L_l^\dagger L_l \otimes I - \frac{1}{2} I \otimes (L_l^\dagger L_l)^T \right). \tag{16}$$

Here $I$ is the identity matrix in the Hilbert space $\mathcal{H}_{\text{phys}}$ and $T$ denotes the transpose of an operator.

### 3.2.2 Representation of an initial state in the purified form

Often we are confronted with the situation that we start a Lindblad evolution from a pure state represented in the MPS formalism, for example this can be the ground state of a Hamiltonian obtained by the MPS ground state search. For this purpose we present the steps of purifying a state $|\psi\rangle$ that is given in MPS form [Eq. 11]. Since the density matrix of this pure state is given by $|\psi\rangle \langle \psi|$, first a copy of the state is created which represents the bra contribution. Then the tensors stemming from the ket and the bra corresponding to the same site are contracted, which results in a set of tensors with six indices

$$\tag{17}$$

In order to obtain single-site tensors for the ket and the bra part, the bond indices are combined, i.e. $(a_{l-1}, a_{l-1}') \to \lambda_{l-1}'$ and $(a_l, a_l') \to \lambda_l'$, before the tensor are separated into the respective single-site parts using a singular value decomposition with regard to the indices

$$|\rho(t+\Delta t)\rangle\!\rangle = e^{\mathcal{L}\Delta t}|\rho(t)\rangle\!\rangle = e^{\mathcal{L}_{\mathrm{odd}}\Delta t/2}e^{\mathcal{L}_{\mathrm{even}}\Delta t}e^{\mathcal{L}_{\mathrm{odd}}\Delta t/2}|\rho(t)\rangle\!\rangle + \mathrm{O}(L\Delta t^3)$$

$$\times\,|\sigma_1\sigma_1'\ldots\sigma_L\sigma_L'\rangle\!\rangle + \mathcal{O}(L\Delta t^3)$$

Figure 2: Dissipative evolution of the purified density matrix in MPS form by a single time step $\Delta t$ using a second order Suzuki-Trotter decomposition for the evolution operator represented by a consecutive application of four-site gates. The two colors mark the affiliation to one of the two parts of the Lindbladian, i.e. either $\mathcal{L}_{\mathrm{odd}}$ or $\mathcal{L}_{\mathrm{even}}$, and the arrow indicates the order of application.

$(\lambda'_{l-1}, \sigma_l) \times (\sigma'_l, \lambda'_l)$. The new bond index, created by the SVD, is denoted by $\lambda_l$, so that the MPS representation of a purified state reads

$$|\rho\rangle\!\rangle = \sum_{\vec{\sigma},\vec{\sigma}'} \sum_{\substack{\lambda_1,\ldots,\lambda_L \\ \lambda'_1,\ldots,\lambda'_{L-1}}} M^{\sigma_1}_{1,\lambda_1} M^{\sigma'_1}_{\lambda_1,\lambda'_1} \ldots M^{\sigma_L}_{\lambda'_{L-1},\lambda_L} M^{\sigma'_L}_{\lambda_L,1} |\sigma_1\sigma_1'\ldots\sigma_L\sigma_L'\rangle\!\rangle. \tag{18}$$

This translates to the following diagrammatic expression

$$|\rho\rangle\!\rangle = \sum_{\substack{\sigma_1\ldots\sigma_L \\ \sigma_1'\ldots\sigma_L'}} \quad \text{} \quad |\sigma_1\sigma_1'\ldots\sigma_L\sigma_L'\rangle\!\rangle\ . \tag{19}$$

It is important to notice that this approach affects the bond dimension drastically. Assuming the bond dimension of the state to purify is given by $D$, the combining of bond indices results in a dimension $D^2$ for the index set $\{\lambda'_l\}$ and the SVD without compression causes an increase to $dD^2$ for the indices $\{\lambda_l\}$. In order to keep the original specified bond dimension $D$, the state to purify needs to be compressed to a bond dimension $\sqrt{D}$ and the spectrum of SVD in the second step needs to be truncated after the $D$ largest values. This poses a very strong constraint, so that particularly strongly entangled states are difficult to purify. Either the need for computational resources increases quadratically or the accuracy, measured by the truncation weight, becomes significantly worse.

### 3.2.3 Time-evolution of the purified density matrix

In analogy to the unitary closed system evolution (Eq. 13), it is also possible to split the Lindbladian into two contributions

$$\mathbb{L} = \mathbb{L}_{\mathrm{odd}} + \mathbb{L}_{\mathrm{even}}, \tag{20}$$

where $\mathbb{L}_{\text{odd}}$ ($\mathbb{L}_{\text{even}}$) covers the ket and bra sites connected by the odd (even) bonds. Again, the terms within $\mathbb{L}_{\text{odd/even}}$ commute, but $[\mathbb{L}_{\text{odd}}, \mathbb{L}_{\text{even}}] \neq 0$. Consequently, the dissipative evolution can also be approximated by the second order Suzuki-Trotter decomposition, which in this case results in a sequence of applications of four-site gates as shown in Fig. 2. This fact raises the challenge, that a gate application causes a stronger increase of the bond dimension

$$
\begin{array}{c}
\xrightarrow{SVD}
\end{array}
\tag{21}
$$

The dimension of the individual indices is marked in blue color, revealing that the bond dimension of a purified state grows from $D$ to up to $d^4 D$ by applying a time evolution gate. This requires a stronger truncation to compress the bond dimension to the original size.

Another important feature is that the use of quantum number conserving codes for the computation of the dissipative evolution is only possible if the Lindbladian is subject to a strong symmetry [67, 68], i.e. not only the Hamilton operator but also every single jump operator of the model respects the conservation law. In the cases considered here, this only applies to $\mathcal{D}_1$ (Eq.4), as the jump operator $S_c^-$ in $\mathcal{D}_2$ (Eq.5) does not conserve the total magnetization.

### 3.2.4 Calculating expectation values within the purification approach

Provided with the time evolved purified density matrix, we are left with the task to calculate expectation values of observables to extract information about the system. The trace relation for the expectation value of an operator $A$ translates to a scalar product

$$
\langle A \rangle = \text{tr}(\rho A) = \langle\!\langle \mathbb{1} | A \otimes I | \rho \rangle\!\rangle, \text{ with } | \mathbb{1} \rangle\!\rangle = \bigotimes_{l=1}^{L} \sum_{\sigma} | \sigma \, \sigma \rangle_l = \bigotimes_{l=1}^{L} \left( | \uparrow\uparrow \rangle_l + | \downarrow\downarrow \rangle_l \right). \tag{22}
$$

Here the first and second spin in $| \sigma \, \sigma \rangle_l$ denote the spin at site $l$ for the ket and the bra part in the purified notation respectively. The state $| \mathbb{1} \rangle\!\rangle$ is the purification of the unnormalized infinite temperature state [69]. Unfortunately, the possibility to be able to encode this state as a product state as shown in Eq. 22 gets lost, when using good quantum numbers, where the vector of the purified density matrix cannot be separated into contributions of single sites anymore. As a result, this state can exhibit a potentially large bond dimension. Another obstacle becomes apparent when exploiting good quantum numbers in the algorithm is that $| \mathbb{1} \rangle\!\rangle$ spans over all symmetry blocks, so that a restriction to a certain quantum number sector makes a manual selection of basis states fulfilling this condition very cumbersome. Here it is helpful to realize, that the purification procedure is subject to a gauge degree of freedom. The measured expectation value is invariant under local unitary transformations of the states [56]. This fact can be used to construct a favorable representation of the bra space of the constituents of $| \mathbb{1} \rangle\!\rangle$, defined by

$$
| \sigma \, \sigma \rangle_l \longrightarrow (I \otimes U) | \sigma \, \sigma \rangle_l. \tag{23}
$$

To this end, the transformation $U$ is chosen such, that the quantum number of the full initial state is equally distributed over local pairs of ket and bra states. For example, if we consider

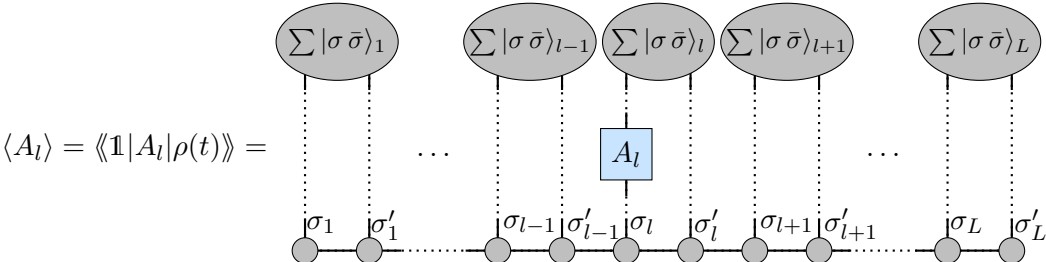

Figure 3: Visualization of the computation of local expectation values using the contraction of the MPS with a local operator $A_l$ and the trace generating state $|\mathbb{1}\rangle$ in the transformed basis (see Eq. 24) represented by a product of spin pairs $\sum_\sigma |\sigma\bar\sigma\rangle_l$ covering associated sites corresponding to the ket and bra part of the density matrix at site $l$. The density matrix is also calculated in the transformed basis as Eq. 25. Note that the observable $A_l$ does not change with this transformation.

the $M_{\text{tot}} = 0$ symmetry sector in the Lindblad model using the dissipator $\mathcal{D}_1$, one possible transformation is the Pauli matrix in $x$-direction $U = \sigma^x$ on the bra sites, so that we can rewrite the state $|\mathbb{1}\rangle\rangle$ as a product of local spin pairs

$$|\mathbb{1}\rangle\rangle \longrightarrow \bigotimes_{l=1}^{L} \sum_\sigma |\sigma\bar\sigma\rangle_l = \bigotimes_{l=1}^{L} (|\uparrow\downarrow\rangle_l + |\downarrow\uparrow\rangle_l) \tag{24}$$

where $\bar\sigma = -\sigma$. This transformation automatically guarantees the restriction of the trace generating state $|\mathbb{1}\rangle\rangle$ to the symmetry sector selected by the initial state. With this, the problems of selecting basis states respecting the quantum number conservation as well as the large bond dimension of an MPS representation of $|\mathbb{1}\rangle\rangle$ are both solved. Also the initial state and the Lindbladian gates need to be adapted to be consistent with this gauge choice. With $\mathbb{U} = \bigotimes_{l=1}^{L} (I \otimes U)$, the transformation relations are

$$
\begin{aligned}
\mathbb{L} &\longrightarrow \mathbb{U}^\dagger \mathbb{L} \mathbb{U} \\
|\rho(t=0)\rangle\rangle &\longrightarrow \mathbb{U}|\rho(t=0)\rangle\rangle.
\end{aligned} \tag{25}
$$

Supposing that the observable can be brought to an efficient tensor representation, we can then proceed to calculate the expectation value. In the exemplary and important case of a local observable the measurement is carried out by the tensor contractions outlined in Fig.3. As explained before only the case with dissipator $\mathcal{D}_1$ conserves the total magnetization. Thus we use this transformation in order to consider the good quantum numbers in the simulation. Applying this transformation to the Lindbladian leaves the Hamiltonian contribution unchanged and only changes the sign of the first part of the dissipator $\mathcal{D}_1$. The observables are calculated in the transformed basis (see Fig. 3).

## 3.3 Monte-Carlo wave function method

Instead of evolving the full density matrix, an alternative way is to compute the evolution of wave function trajectories in the original Hilbert space [3, 36, 37]. This is at the expense of the need for a sampling over many realizations due to the presence of stochastic processes,

originating from the action of the environment on the system. This approach, known as the unraveling of the master equation, can be realized by piece-wise deterministic jump processes, where the deterministic time evolution of a state is interrupted by the application of jump operators. The deterministic evolution is performed with regard to an effective Hamiltonian

$$H_{\text{eff}} = H_{\text{XXZ}} - \frac{i\hbar\gamma}{2} \sum_l L_l^\dagger L_l, \tag{26}$$

where $\{L_l\}$ are the the jump operators of the respective model, i.e. $\{L_l\} = \{S_j^z, j = 1, 2, \ldots, L\}$ for $\mathcal{D}_1$ and $\{L_l\} = \{S_c^-, c = \text{central site}\}$ for $\mathcal{D}_2$. As this effective Hamiltonian is not Hermitian, the corresponding evolution is non-unitary, resulting in a decay of the norm of the state over time. The creation of a single time-evolved trajectory sample can be summarized as follows:

1. Define the initial state, which is either a pure state or is selected according to the probability weights in a mixture of states.

2. Draw a random number $\eta \in [0, 1)$.

3. Evolve the state under $H_{\text{eff}}$ by a sequence of small time steps $\Delta t$ until $\langle \psi(t) | \psi(t) \rangle \leq \eta$.

4. (a) Draw a jump operator $L_{l'}$ according to the probability distribution $\frac{p_{l'}}{\sum_l p_l}$, with

$$p_l = \Gamma_l \langle \psi(t) | L_l^\dagger L_l | \psi(t) \rangle, \tag{27}$$

   (b) apply the selected operator $L_{l'}$ to the state and renormalize it

$$|\psi(t)\rangle = \frac{L_{l'} |\psi(t)\rangle}{\sqrt{\langle \psi(t) | L_{l'}^\dagger L_{l'} |\psi(t)\rangle}}. \tag{28}$$

5. Iterate from 2 until the final time is reached.

Performing the average over many trajectories generated by this scheme ultimately yields an estimate of the density matrix that is accurate up to the first order in the time step. The density matrix can be approximated by a Monte-Carlo (MC) average over a finite number $R$ of trajectory samples $|\psi_r(t)\rangle$

$$\rho \approx \frac{1}{R} \sum_{r=1}^R |\psi_r(t)\rangle \langle \psi_r(t)| \tag{29}$$

giving the technique name of Monte-Carlo wave function (MCWF) method. Similarly, the expectation values of observables can also be evaluated as

$$\langle\!\langle \hat{O}(t) \rangle\!\rangle \approx \frac{1}{R} \sum_{r=1}^R \langle \psi_r(t)| \hat{O} |\psi_r(t)\rangle, \tag{30}$$

where $\langle\!\langle \ldots \rangle\!\rangle$ denotes the MC average and the stochastic accuracy of the independent sampling approach is given by the time-dependent standard deviation of the mean

$$\sigma_{\text{mean}}(\hat{O}(t)) = \sqrt{\frac{1}{R(R-1)} \sum_{r=1}^R \left( \langle \psi_r(t)| \hat{O} |\psi_r(t)\rangle - \langle\!\langle \hat{O}(t) \rangle\!\rangle \right)^2}. \tag{31}$$

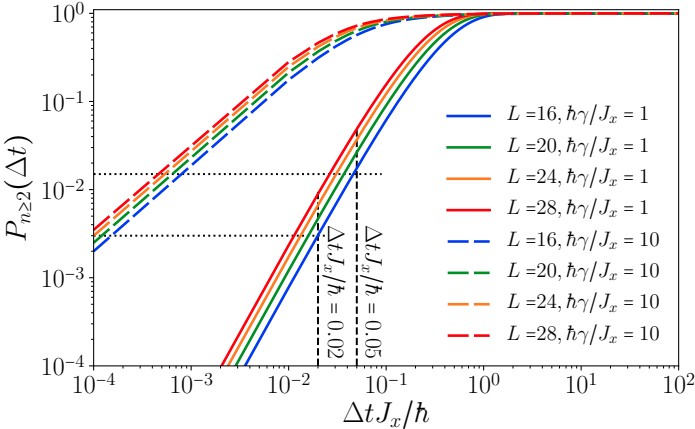

Figure 4: Probability distribution for two or more jump events happening in the same time step to estimate the necessary time step for the model with dissipator $\mathcal{D}_1$ and different system sizes and dissipation strengths. The vertical dashed lines mark the time-steps typically chosen in the calculations and the horizontal dotted lines are the corresponding probability for $L = 16$.

The random sampling of an application time in steps 2 and 3 of the algorithm is equivalent to drawing a waiting time $\tau$ until the next jump occurs according to the distribution

$$P(t,\tau) = 1 - \langle\psi(t)|e^{iH_{\text{eff}}\tau/\hbar}e^{-iH_{\text{eff}}\tau/\hbar}|\psi(t)\rangle. \tag{32}$$

This quantity is important for the convergence of the method as it can be used to determine a sufficiently small time step when simulating the non-unitary evolution under the effective Hamiltonian. It is important to choose a time step small enough so that the probability of more than one jump event happening in the span of a time step is negligible. For the system with dissipator $\mathcal{D}_1$ the imaginary part of the effective Hamiltonian (Eq. 26) is equal to $-\frac{\gamma}{2}\sum_l S_l^z S_l^z = -\frac{\gamma L}{8}$. Thus the waiting time distribution (Eq. 32) simplifies to

$$P(t,\tau) = 1 - e^{-\gamma L\tau/4}. \tag{33}$$

The waiting time distribution is the probability of having a jump in the interval $[t, t+\tau]$ which is independent of the initial time $t$ here.

The probability to have two or more jumps in this interval is $P_{n>2}(\tau) = 1 - e^{-\gamma L\tau/4} - \frac{\gamma L\tau}{4}e^{-\gamma L\tau/4}$. In Fig. 4 we show the probability of two or more jumps taking place in one time step for the model with dissipator $\mathcal{D}_1$. When a good value for $\Delta t$ has been found for one parameter configuration, the dissipation strength and the system size are crucial for estimating a suitable time step for another set-up.

For the determination of a single trajectory applying the introduced $t$MPS algorithm Eq. 13 offers a promising solution for computing the deterministic part of the evolution (see [70] and references therein). Using this procedure, we can access the dissipative time evolution of a spin-1/2 chain under the action of the Lindbladian. It is noteworthy that here the use of symmetries to reduce the computational complexity only requires the conservation of the associated quantum numbers by the effective Hamiltonian. Consequently, in contrast to the purification approach, quantum number conserving codes can be used for the simulation of both dissipators $\mathcal{D}_1$ and $\mathcal{D}_2$. In Fig. 5 we present for the example of the dissipator $\mathcal{D}_1$ the

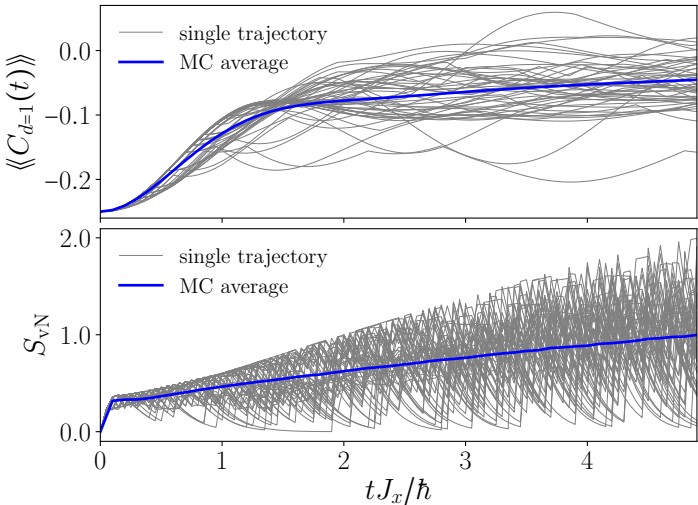

Figure 5: Dissipative evolution of local equal-time correlation functions $C_d(t)$ (top) and the von Neumann entropy $S_{\mathrm{vN}}$ (bottom) of a spin-1/2 chain initially prepared in the Neel state under the effect of bulk dephasing $\mathcal{D}_1$. We show both, the Monte-Carlo average with statistical error bars (not visible, since they are below the line width) and the measured values for a selection of stochastically sampled trajectories for a chain of length $L = 32$, dissipation strength $\hbar\gamma/J_x = 1$, spin anisotropy $J_z/J_x = 1$, a time step $\hbar\Delta t/J_x = 0.05$, a truncation goal $\varepsilon = 10^{-12}$, a maximal bond dimension $D = 100$ and $10^4$ samples for the MC average.

evolution of the local equal-time correlation function

$$C_d(t) = \langle S_{L/2}^z S_{L/2+d}^z \rangle(t), \tag{34}$$

and the von Neumann entropy $S_{\mathrm{vN}}$ starting from an alternating spin configuration $|\psi_{\mathrm{Neel}}\rangle = |\uparrow\downarrow\uparrow\downarrow\ldots\rangle$, known as the classical Neel state. For both quantities we show the Monte-Carlo average as well as a selection of measurements of individual trajectory samples. Within the MPS, the von Neumann entropy is connected to the required maximally allowed bond dimension in order to obtain a good description of the state. Here, we find that during the evolution a spreading around the mean develops so that the maximum entanglement in individual samples can be significantly larger than the mean value suggests. The relationship between entropy and the necessary bond dimension is highlighted further for a single trajectory created from the same random seed for different maximal bond dimensions $D$ in Fig. 6. It is evident that calculations with a fixed maximum value for the bond dimension can only provide a sufficiently accurate representation of the entanglement up to a certain time.

## 4    Methods for the computation of two-time correlation functions in open quantum systems

We will discuss the different concepts for determining two-time correlation functions in open quantum systems with the methods introduced in Sec. 3, which will later form the basis for

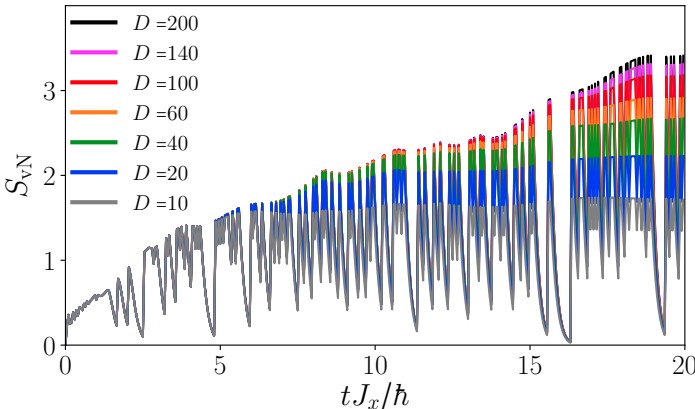

Figure 6: Evolution of the von Neumann entropy $S_{\mathrm{vN}}$ of a single trajectory sample with different maximal values for the bond dimension for a chain of length $L = 32$, dissipation strength $\hbar\gamma/J_x = 1$, spin anisotropy $J_z/J_x = 1$, time step $\hbar\Delta t/J_x = 0.05$ and truncation goal $\varepsilon = 10^{-12}$.

the comparison. More precisely, the goal is to compute correlation functions of the form

$$g(t_2, t_1) = \langle B(t_2)A(t_1)\rangle, \tag{35}$$

where the two operators are applied at different times during the course of a dissipative time evolution.

## 4.1  Two-time correlations within the purification approach

The purification approach can be straight-forwardly extended in order to determine two time correlation functions [1]. After purifying the initial state, the reshaped density matrix is evolved until time $t_1$, where the operator $A$ is applied, followed by an evolution to $t_2$, where an ordinary measurement of $B$ is carried out

$$\langle B(t_2)A(t_1)\rangle = \langle\!\langle \mathbb{1}| \, (B \otimes I) \, \mathrm{e}^{\mathbb{L}(t_2 - t_1)} \, (A \otimes I) \, \mathrm{e}^{\mathbb{L}t_1} |\rho(t = 0)\rangle\!\rangle. \tag{36}$$

Only a single time evolution needs to be calculated. Nevertheless, the fact of the evolution taking place in a doubled Hilbert space, can potentially result in the need for a large bond dimension.

## 4.2  Stochastic sampling: two approaches to two-time correlations

In the following we introduce two different approaches to evaluate two-time correlation functions in the framework of Monte-Carlo wave functions. The first step is to evolve an initially prepared state, which is in the case of a mixed state drawn according to the weights in the density matrix, up to the application time $t_1$ of the first operator. This is performed using the unraveling scheme of piece-wise deterministic jump processes. For each trajectory one defines the state $|\phi(t_1)\rangle \equiv A|\psi(t_1)\rangle$. In principle it seems that we are left with the task of calculating the dissipative time-evolution of the two states $|\phi(t_1)\rangle$ and $|\psi(t_1)\rangle$ up to time $t_2$ and then the expectation value with the operator $B$, i.e.

$$g_2(t_2, t_1) = \langle\psi(t_2)| \, B \, |\phi(t_2)\rangle. \tag{37}$$

split trajectories    joint jumps $L_j$

Figure 7: Sketch of the creation of one sample for the computation of two-time correlators following [2]. After the evolution up to $t_1$ the trajectory is copied, followed by a time span characterized by an independent deterministic non-unitary evolution interrupted by joint jump operator applications.

However, while this expression is well-defined for states of a closed system, the transfer to a stochastic sampling approached is more involved. In the next two sections we outline two methods which are well defined for the stochastic sampling approach. Subsequently, we compare the efficiency of both concepts.

### 4.2.1   Joint evolution of two states following Breuer et al. [2]

The first idea, developed by Breuer et al. [2], uses a doubling of the Hilbert space at time $t_1$ by introducing for each trajectory the vector

$$|\Theta(t_1)\rangle = \begin{pmatrix} |\psi(t_1)\rangle \\ |\phi(t_1)\rangle \end{pmatrix}. \tag{38}$$

As the matrix $\tilde{\rho}(t_1) = |\Theta(t_1)\rangle \langle \Theta(t_1)|$ again fulfills all properties of a physical density matrix, the task is reduced to recover the time dependence in Eq. 37. By defining a new Hamiltonian operator and new jump operators acting on the doubled space

$$\tilde{H} = \begin{pmatrix} H & 0 \\ 0 & H \end{pmatrix} \text{ and } \tilde{L}_l = \begin{pmatrix} L_l & 0 \\ 0 & L_l \end{pmatrix}, \tag{39}$$

and using this in a Lindblad-type equation with Hamiltonian $\tilde{H}$ and jump operators $\tilde{L}_l$, we arrive at Lindblad equations for all matrix blocks [2]. As a result, we can apply the same unraveling approach as before on the doubled space to compute two-time correlation functions. Since the operators do not couple the two subspaces, a separate time evolution of $|\psi(t)\rangle$ and $|\phi(t)\rangle$ is possible, where only the application time and the selection of jump operators is determined based on the joint evolution (see Fig. 7). The algorithm to create a single trajectory can be condensed to the following steps:

1. Initialize the wave function $|\psi(t_0)\rangle$ in the original Hilbert space and evolve it until $t_1$ using the introduced piece-wise deterministic process.

2. Make a copy of the state at time $t_1$, apply the operator $A$ to it $|\phi(t_1)\rangle \equiv A |\psi(t_1)\rangle$ and define a state in the doubled space $|\Theta(t_1)\rangle \equiv (|\psi(t_1)\rangle, |\phi(t_1)\rangle)^T$, where $T$ denotes the transpose. To ensures the transfer of accumulated jump probability at $t_1$ to the doubled space we normalize this state to $|\tilde{\Theta}(t_1)\rangle \equiv \frac{1}{\sqrt{\Omega}} |\Theta(t_1)\rangle \equiv (|\tilde{\psi}(t_1)\rangle, |\tilde{\phi}(t_1)\rangle)^T$ with the normalization factor $\Omega = \langle \Theta(t_1)| \Theta(t_1)\rangle / \langle \psi(t_1)| \psi(t_1)\rangle$. Here the norm of the new state is the same as of the initial state ($\langle \tilde{\Theta}(t_1)| \tilde{\Theta}(t_1)\rangle = \langle \psi(t_1)| \psi(t_1)\rangle$).

3. Evolve both states independently under the effective non-Hermitian Hamiltonian while sampling jumps simultaneously according to the joint loss of norm of $|\tilde{\Theta}(t)\rangle$.

4. Repeat the procedure until the time $t_2$ is reached where one obtains $|\tilde{\Theta}(t_2)\rangle = \begin{pmatrix} |\tilde{\psi}(t_2)\rangle \\ |\tilde{\phi}(t_2)\rangle \end{pmatrix}$.

   Then, use the components of this state in order to measure the two-time correlation function by calculating the full overlap

$$\langle B(t_2)A(t_1)\rangle = \frac{\langle\tilde{\psi}(t_2)|\,B\,|\tilde{\phi}(t_2)\rangle}{\langle\tilde{\Theta}(t_2)|\,\tilde{\Theta}(t_2)\rangle}\Omega.$$

This method can be further extended to multi-time correlation functions by additional operator applications between the first and the final application time [2].

### 4.2.2  Separate evolution of four trajectories following Mølmer et al. [3]

An alternative strategy, established by Mølmer et al. [3], uses the quantum regression theorem [39, 71] to represent the time dependence of two-time correlation functions in terms of the evolution of quantities, which only involve equal-time measurements. The proof relies on the quantum regression theorem which states that if the evolution of equal-time observables for the set of operators $\{B_i\}$ is described by a closed set of differential equations

$$\frac{\mathrm{d}}{\mathrm{d}t_1}\langle B_i(t_1)\rangle = \sum_j G_{ij}\langle B_j(t_1)\rangle, \tag{40}$$

the two-time correlations are generated by the same kernel $G$ as

$$\frac{\mathrm{d}}{\mathrm{d}(t_2 - t_1)}\langle B_i(t_2)A(t_1)\rangle = \sum_j G_{ij}\langle B_j(t_2)A(t_1)\rangle. \tag{41}$$

The idea of this approach relies on constructing four new states at time $t_1$ for each trajectory such that a combination of the expectation values of the operator applied at $t_2$ for each of these state gives the corresponding two-time correlation. To this end, the four states are defined by applying the first operator $A$ [3],

$$|\chi_R^\pm(t_1)\rangle = \frac{1}{\sqrt{\mu_R^\pm}}(I \pm A)\,|\psi(t_1)\rangle\,,$$

$$|\chi_I^\pm(t_1)\rangle = \frac{1}{\sqrt{\mu_I^\pm}}(I \pm iA)\,|\psi(t_1)\rangle\,, \tag{42}$$

where for each trajectory $|\psi(t_1)\rangle$ the state at $t_1$ is obtained by the usual unraveling scheme. Further, $\mu_R^\pm$ and $\mu_I^\pm$ normalize the respective states. These new states evolve independently from time $t_1$ to $t_2$ with the same unraveling scheme (see Fig. 8). By properly combining the expectation values of the second operator $B$ for each state the two-time correlation functions can be recovered [3]

$$\langle B(t_2)A(t_1)\rangle = \frac{1}{4}\left[\mu_R^+\langle\chi_R^+(t_2)|\,B\,|\chi_R^+(t_2)\rangle - \mu_R^-\langle\chi_R^-(t_2)|\,B\,|\chi_R^-(t_2)\rangle\right.$$
$$\left. - i\mu_I^+\langle\chi_I^+(t_2)|\,B\,|\chi_I^+(t_2)\rangle + i\mu_I^-\langle\chi_I^-(t_2)|\,B\,|\chi_I^-(t_2)\rangle\right]. \tag{43}$$

Consequently, it is possible to access the two-time correlation functions by evolving the four states from Eq. 42 separately. In contrast to the MCWF technique of the doubled Hilbert

split trajectories    independent evolution

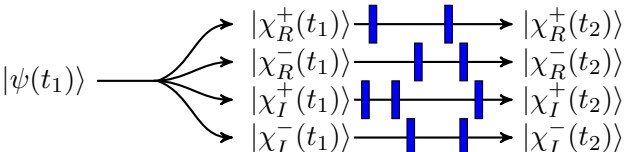

Figure 8: Sketch of the creation of one sample for the computation of two-time correlators following Ref. [3]. The initial dynamics up to time $t_1$ is followed by the splitting of the wave function into four trajectories from which the two-time expectation value at $t_1$ can be reconstructed. Then each sub-trajectory is evolved separately.

space, also the jump sampling procedure is completely independent so that it is possible to compute the four trajectories after $t_1$ in parallel. In addition to the average over the four states, the average over many trajectories from the initial time to $t_1$ and from time $t_1$ to $t_2$ needs to be taken. A disadvantage of this approach using four different state is, however, that it does not offer a straight-forward extension to multi-time correlation functions.

There are two special cases, defined by the properties of the operators $A$ and $B$ of the correlation function. If the operator at time $t_1$ is Hermitian ($A^\dagger = A$) and commutes with the operator at $t_2$ ($[A, B] = 0$), it is possible to consider two instead of four trajectories and to calculate the two-time correlation as

$$\langle B(t_2)A(t_1)\rangle = \frac{1}{4}\left[\mu_R^+\langle\chi_R^+(t_2)|B|\chi_R^+(t_2)\rangle - \mu_R^-\langle\chi_R^-(t_2)|B|\chi_R^-(t_2)\rangle\right]. \tag{44}$$

To ensure that the comparison between the two approaches is as fair as possible, we concentrate on such a special case for the two-time correlator in Eq. 45, since for this case the number of trajectories needed in both approaches is equal. We expect that in the more general case where even four trajectories are needed in the approach by Mølmer et al., this approach will become even more costly. The second case is more subtle and appears in the context of conserved quantum numbers. If the Lindbladian evolution allows the partition of the super-operator into symmetry blocks, and the operator $A$ at $t_1$ couples different blocks, the resulting states $|\chi_{R,I}^\pm\rangle$ cannot longer be addressed to a single block. Instead it is possible to evolve the parts of the different sectors separately as they are decoupled for all times $t > t_1$. A closer look reveals that in this case the introduced scheme is equivalent to the approach by Breuer et al. from the last section.

## 5 Comparison of the different methods to determine two-time correlations functions in open quantum systems

In this section we compare the performance of the different methods to calculate two-time correlations using the spin model described in section 2. We focus on the two-time correlation function relating two applications of local $S^z$-operators at sites separated by a distance $d$ given by

$$C_d(t_2, t_1) = \langle S_c^z(t_2)S_{c+d}^z(t_1)\rangle. \tag{45}$$

Recall, $c$ is the central site for a chain with an odd number of sites and the center-left site otherwise. This correlation function has been proven to be essential to uncover interesting

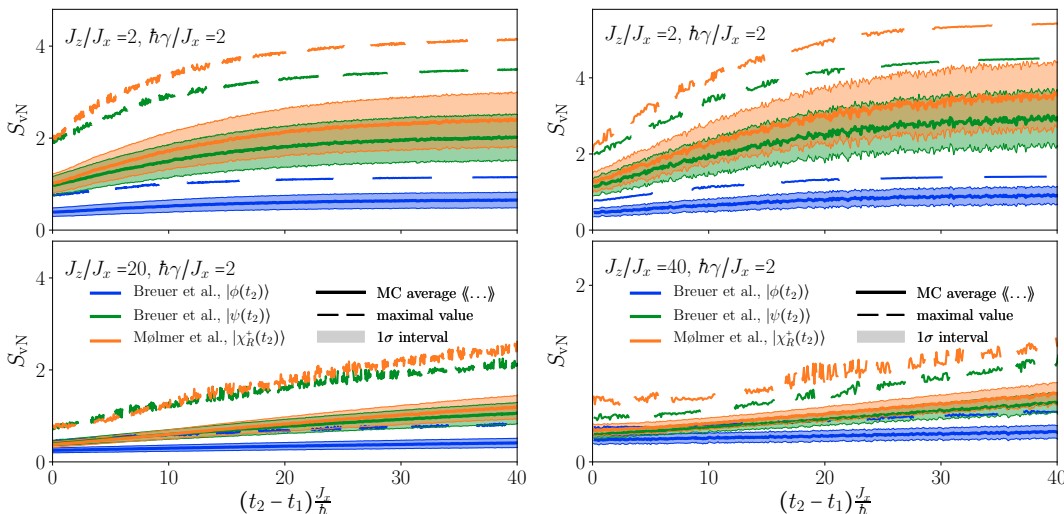

Figure 9: Evolution of the von Neumann entropy as a resource measure for different branches of Breuer and Mølmer MCWF approach to computing two-time correlation functions for two different sets of model parameter. Results are shown for $t_1 J_x/\hbar = 5$, a time step of $\Delta t J_x/\hbar = 0.02$ and $10^4$ trajectory samples for (left panel) an exact MPS representation with chains of length $L = 14$ and (right panel) non-exact MPS representation for $L = 20$.

dynamical regimes displaying physical phenomena such as aging or hierarchical dynamics [1]. We start by the comparison of the two stochastic approaches for the dissipator $\mathcal{D}_1$ in subsection 5.1. We use as an initial state a classical Neel state. We find that typically the Breuer et al. approach combined with the MPS methods performs better than the Mølmer et al. approach. We continue in comparing the better performing stochastic approach, the approach by Breuer et al., to the purification approach in subsection 5.2. For the considered situation the purification approach greatly outperforms the stochastic approach. Reasons of the excellent performance of the purification approach in this situation are the 'easy' initial state and the conservation of the magnetization by the Lindblad dynamics and most importantly, the low matrix dimension needed.

In subsection 5.3 we turn to the local dissipator $\mathcal{D}_2$. We consider the situation where the initial state is the ground state of the Hamiltonian and the dissipation is switched on at time $t = 0$. We compare again the stochastic approach of Breuer et al. to the purification approach. The entangled initial state and the absence of the conservation of magnetization constitutes an additional difficulty for the purification approach. We find that for most parameters considered the stochastic approach is much more suited to treat this situation efficiently.

## 5.1 Comparison of stochastic approaches for the dephasing noise $\mathcal{D}_1$

In this section we compare the efficiency of the two stochastic approaches in order to calculate two-time correlation functions.

We calculate the correlation function from Eq. 45 for the spin model with the dissipator $\mathcal{D}_1$. The choice for the correlation functions allows us to only use two states, i.e. $|\chi_R^\pm\rangle$, in the method by Mølmer et al.

First, we investigate the cost of generating a single two-time trajectory sample with MPS

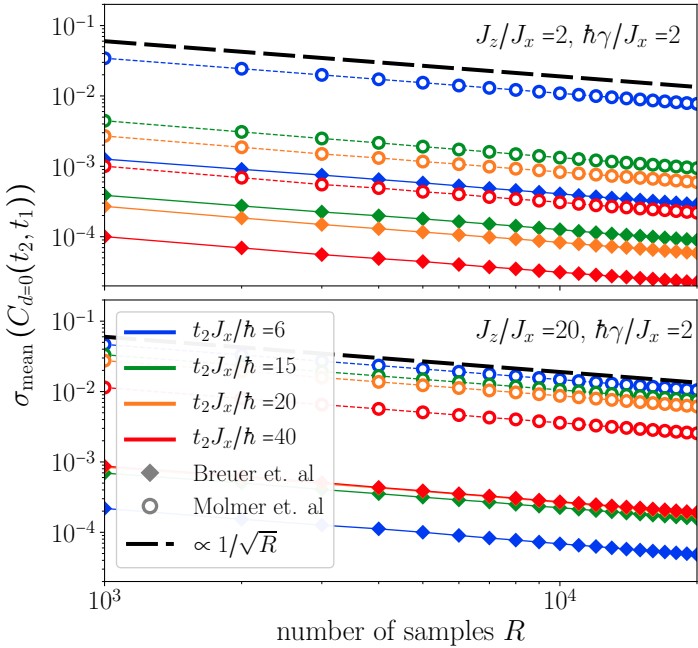

Figure 10: Standard deviation of the mean of the two-time correlation function in dependence of the number of Monte-Carlo samples $R$ measured at four different times $t_2 > t_1$. The black dashed line is a guide to the eye and data is shown for the interaction strengths $J_z/J_x = 2$ (top panel) and $J_z/J_x = 20$ (bottom panel), $L = 14$, $\hbar\gamma/J_x = 2$, $t_1 J_x/\hbar = 5$, an exact MPS representation and a time step of $\Delta t J_x/\hbar = 0.02$.

methods. To this end, the entanglement entropy is evaluated for all times $t > t_1$. As it is directly linked to the needed bond dimension, it serves as an architecture-free measure of the required resources in terms of run time and memory consumption. In Fig. 9 we show the von Neumann entropy for the two branches in the Breuer approach, where $|\phi(t_1)\rangle = A|\psi(t_1)\rangle$, as well as for the $|\chi_R^{\pm}\rangle$ branches of the Mølmer procedure for different parameter sets and system sizes. As the results for the two Mølmer branches are indistinguishable in the plot, we only present data for $|\chi_R^{+}\rangle$. Besides the MC average we also show the maximum measured entropy over all sampled trajectories and the $1\sigma$ interval of the measured standard deviation. In the left panel we use an exact MPS representation for $L = 14$ sites in order to avoid biases introduced by the truncation. We see that the entropy increases over time and the statistical deviation increases in all cases. However, for both parameter sets presented it is evident that the two branches of the Breuer approach generate significantly less entropy than the Mølmer branches. Furthermore, a strong dependence on the model parameters exists. The von Neumann entropy is much smaller for large anisotropy parameters $J_z/J_x$. In the right panel of Fig. 9 larger system size $L = 20$ are shown. The total von Neumann entropy generated ($J_z/J_x = 2$) is larger for larger system sizes. However, concerning the behavior of the different approaches the findings of the small system size are confirmed that the von Neumann entropy in the Mølmer approach grows a bit faster than the one of the approach by Breuer et al.. Another important factor is the convergence of the Monte-Carlo averages with respect to the number of samples. We present the scaling of the standard deviation of the mean with the number of samples at different time points in Fig. 10. As the sampling is

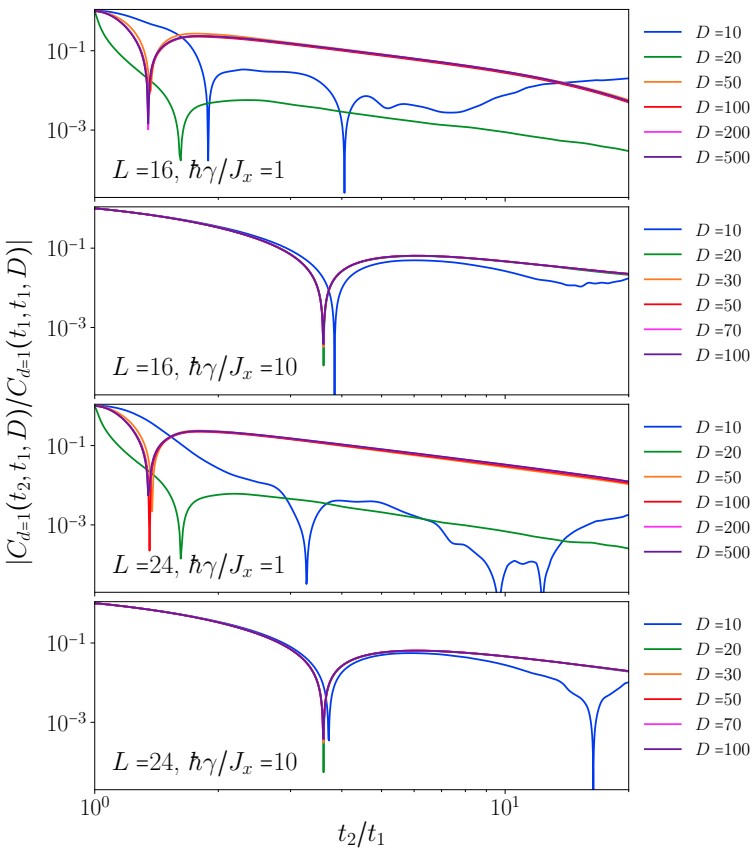

Figure 11: Convergence of the purification approach for spin-$1/2$ chains of different length $L$, different dissipation strengths and $J_z/J_x = 2$. We chose a vanishing truncation goal $\varepsilon = 0$ to enforce the realization of the bond dimension $D$ and use a time step $\Delta t J_x/\hbar = 0.05$ and $t_1 J_x/\hbar = 5$.

statistically independent we observe an inverse square root scaling in all cases. However, this is more than one order of magnitude smaller for the joint evolution suggested by Breuer et al. than for the evolution of Mølmer et al.. In addition, the standard deviation is smaller at later times, which indicates the approach of the unique infinite temperature steady state.

In summary of this section, we can conclude, that for the specific model and parameter sets considered here, the approach of Breuer et al. is favorable over the one by Mølmer et al. in terms of both, memory and run time.

## 5.2 Comparison of Monte-Carlo wave function and purification method for $\mathcal{D}_1$

After having identified the approach by Breuer et al. as superior compared to Mølmer et al. for the considered situation, we compare this approach to the purification method. Since the exact increase of the bond dimension is unknown in both cases, the trade-off between the larger Hilbert space and the cost of sampling many trajectories needs to be evaluated carefully.

We compare the accuracy of the two-time correlation function $C_d(t_2, t_1)$ normalized to

the value at $t_1$ during the dissipative evolution of a spin-1/2 chain initially prepared in the Neel state with bulk dephasing, i.e. using the dissipator $\mathcal{D}_1$. We previously performed such calculations in Ref. [1] and analyzed there the time step required to obtain a good accuracy. In the following we use these time steps for our comparison. Fig. 11 shows the results obtained by the purification method with different values for the bond dimension. This method converges faster in terms of the bond dimension for larger values of the dissipation strength and smaller system sizes. Since it is not clear at which time the greatest inaccuracy exists, we use the maximum deviation of the normalized two-time correlations from the results of the largest achievable bond dimension $D_{\max}$ over the entire time as a measure for the error

$$\Delta_{\max}(D) = \max_{t_2} \left\{ \left| \frac{C_1(t_2, t_1, D)}{C_1(t_1, t_1, D)} - \frac{C_1(t_2, t_1, D_{\max})}{C_1(t_1, t_1, D_{\max})} \right| \right\}. \tag{46}$$

The two-time correlation $C_1(t_2, t_1)$ which is calculated using bond dimension $D$ is denoted by $C_1(t_2, t_1, D)$. Let us note that this measure for the error not only measures the 'pure' truncation error, since the trajectories with a different bond dimension also have a different stochastic nature.

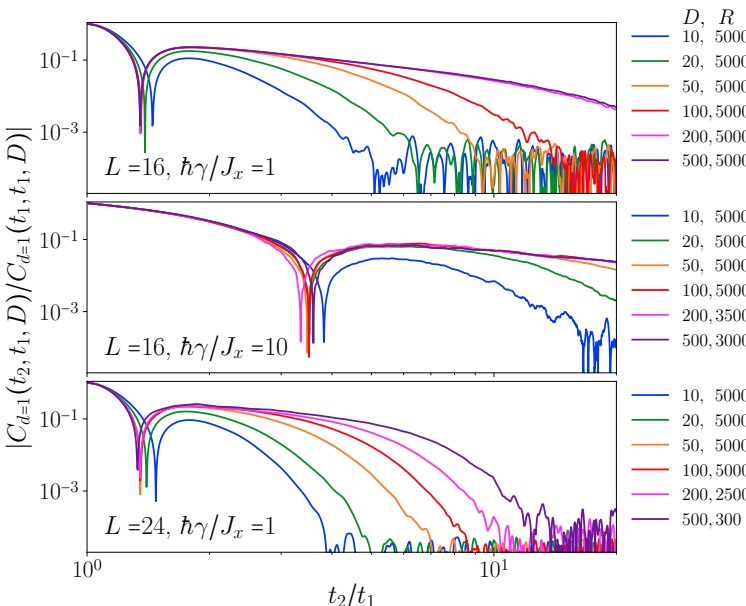

Figure 12: Convergence of two-time correlations obtained from MCWF using the method by Breuer et al. for different system sizes $L$ and dissipation strengths and a spin interaction anisotropy $J_z/J_x = 2$ with $t_1 J_x/\hbar = 5$. Here we use the truncation goal $\varepsilon = 0$ and a time step $\Delta t J_x/\hbar = 0.02$ for $\hbar\gamma/J_x = 1$ and $\Delta t J_x/\hbar = 0.002$ for $\hbar\gamma/J_x = 10$. The bond dimension $D$ and the number of MC samples $R$ are given for each curve.

The results for the stochastic sampling approach presented in Fig. 12 point in a similar direction, i.e. stronger dissipation and smaller systems require a smaller bond dimension to yield reliable results. Surprisingly, the convergence of the stochastic approach with increasing the bond dimension is generally much slower compared to the purification method. In addition, the accuracy is also influenced by the number of stochastic samples taken. To capture the error from the stochastic sampling, we first define the maximum value of the standard

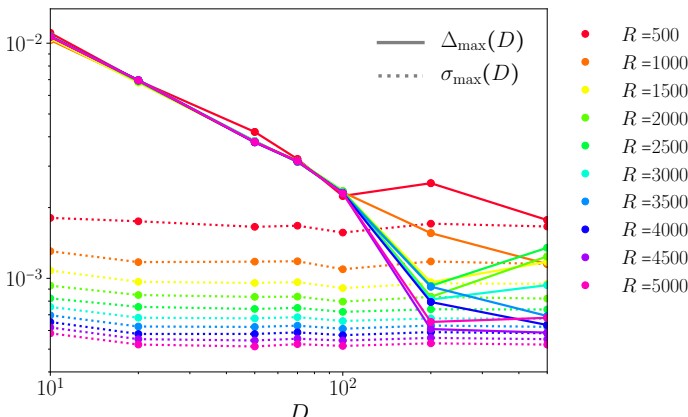

Figure 13: Error estimation of the MCWF computation of the normalized two-time correlator $C_{d=1}(t_2, t_1)$ for $L = 16$ sites, $\hbar\gamma/J_x = 1$, $J_z/J_x = 2$, $t_1 J_x/\hbar = 5$ and a time step of $\Delta t J_x/\hbar = 0.02$. The solid lines show the MPS truncation error $\Delta_{\max}$ and the dashed lines show the statistical error $\sigma_{\max}$ for different number of trajectories $R$. The statistical error becomes dominant when choosing the bond dimension sufficiently large. Here $\Delta_{\max}$ is calculated with respect to the results with $D = 500$ and $R = 10^4$.

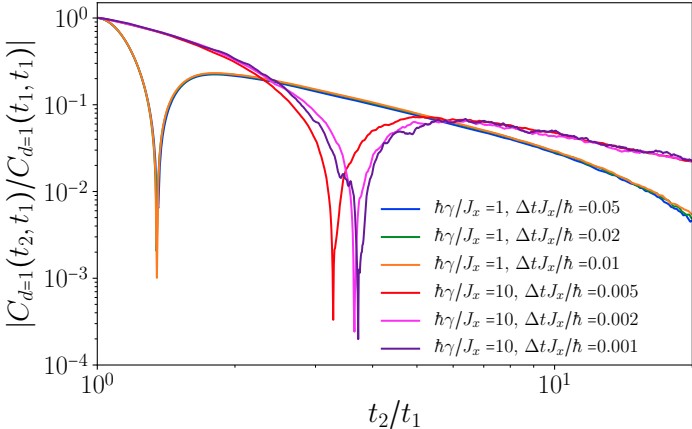

Figure 14: Time step convergence of MCWF approach for different dissipation strengths for a chain of $L = 16$ spins, $J_z/J_x = 2$, $t_1 J_x/\hbar = 5$, truncation goal $\varepsilon = 0$ and a bond dimension $D = 500$. The sample size is $10^4$.

deviation of the mean in time as

$$\sigma_{\max}(D) = \max_{t_2}\left\{\sigma_{\text{mean}}\left(\frac{C_1(t_2, t_1, D)}{C_1(t_1, t_1, D)}\right)\right\} \tag{47}$$

and then the total error for the MCWF approach as the maximum of $\Delta_{\max}(D)$ and $\sigma_{\max}(D)$. As can be seen in Fig. 13, increasing the number of samples used, the sampling error decreases. Further, with increasing bond dimensions, the truncation error becomes less important and $\Delta_{\max}(D)$ becomes of the same order as $\sigma_{\max}(D)$.

Another important fact to consider is the dependence of the choice for the time step on

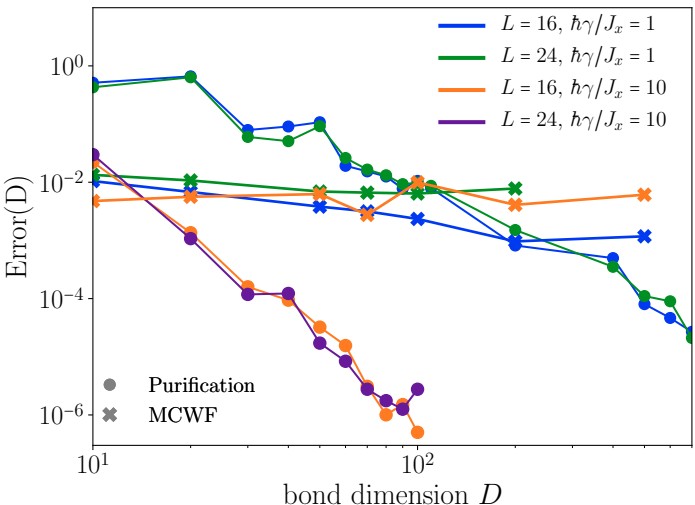

Figure 15: Comparison of the two-time correlation functions error (Error($D$)) calculated in the purification method and the MCWF approach introduced by Breuer et al. for $J_z/J_x = 2$ and $t_1 J_x/\hbar = 5$. For the purification data a time step of $\Delta t J_x/\hbar = 0.05$ has been used and for the MCWF method $\Delta t J_x/\hbar = 0.02$ for $\hbar\gamma/J_x = 1$ and $\Delta t J_x/\hbar = 0.002$ for $\hbar\gamma/J_x = 10$. As a reference to calculate the $\Delta_{\max}$ used in Error($D$) we choose $D_{\max} = 900$ for $\hbar\gamma/J_x = 1$ and $D_{\max} = 110$ for $\hbar\gamma/J_x = 10$ for the purification method. For the MCWF approach we consider $D_{\max} = 1000$ with $R = 10^4$ for $L = 16$ and $\hbar\gamma/J_x = 1$, $D_{\max} = 1000$ with $R = 2500$ for $L = 16$ and $\hbar\gamma/J_x = 10$ and $D_{\max} = 500$ with $R = 300$ for $L = 24$ and $\hbar\gamma/J_x = 1$.

the dissipation strength and the system size in the MCWF approach as mentioned in the discussion of Fig. 4. Fig. 14 confirms that $\Delta t J_x/\hbar = 0.02$ is a suitable time step for the dissipation strength $\hbar\gamma/J_x = 1$. Using the extrapolation indicated in Fig. 4 we estimate $\Delta t J_x/\hbar \approx 0.002$ as a potential step size for $\hbar\gamma/J_x = 10$. However, the convergence analysis reveals that this is still not sufficient to obtain high quality simulation results. We find that larger dissipation strengths require a very small time step for MCWF simulations in this cases, which causes substantially longer run times and additionally increases the error caused by the MPS truncations, as more gate application with subsequent singular value decompositions are needed to reach a certain simulation time.

With the introduced benchmark strategy, i.e. using

$$\text{Error}(D) = \begin{cases} \Delta_{\max}(D) & : \text{purification} \\ \max\{\Delta_{\max}(D), \sigma_{\max}(D)\} & : \text{MCWF} \end{cases} \qquad (48)$$

as an error measure for the two approaches, it is now possible to compare them quantitatively. The dependence of the error on the bond dimension is presented in Fig. 15 for two different system sizes and dissipation strengths. While a weak dependence on the system size is noticeable, the dissipation strength turns out to be the decisive parameter. For the purification calculations the convergence regarding the bond dimension is much faster for strong dissipation. The accuracy of the stochastic sampling is relatively quickly dominated by the statistic error, so that the behavior with the bond dimension appears to be almost constant. Consequently, there are crossing points above which the accuracy of the purification method is better for the same bond dimension.

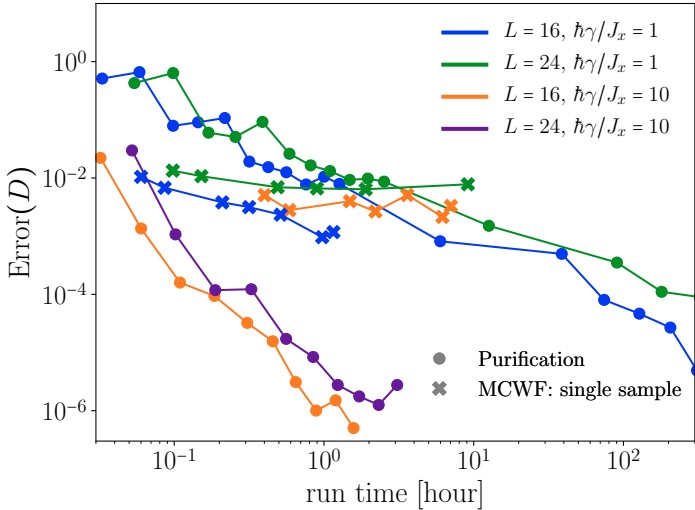

Figure 16: Run time measurement to achieve certain numerical accuracy provided by the measured error of the two-time correlation functions using machines with clock frequency 2.6 GHz. The run time of the creation of single Monte-Carlo sample is compared to the full run time of a purification based computation. Parameters are the same as in Fig. 15.

To put these results into the context of realistic numerical resources, we show in Fig. 16 the required run time for a certain error on a computer cluster consisting of machines with 2.6 GHz clock frequency and sufficient memory resources. In case of strong dissipation, the generation of a single trajectory already takes longer than the evolution using the purification of the reshaped density matrix due to the necessity of a much smaller time step. For weaker dissipation strengths the run time of a single trajectory becomes comparable to the purification approach. However, the total run time of the stochastic sampling computation, requiring a sample size of the order $10^4$ trajectories, is orders of magnitude larger than the full evolution, even with a massive parallelization of the sampling process. This means that the purification approach in these cases is strongly favorable over the stochastic approach.

## 5.3 Comparison of purification and stochastic approach for local dissipation $\mathcal{D}_2$

To demonstrate the strong influence of the model and the initial state on the method choice, we continue by supplementing the findings of the last chapter with investigating results of another set-up. For this purpose, we turn to the dissipator $\mathcal{D}_2$, which only contains one jump operator given by the spin lowering operator $S_c^-$ acting on the central site of a system with an odd number of spins. As this jump operator violates the conservation of the total magnetization during the Lindbladian evolution, it is not possible to exploit symmetry properties in the evolution of the purified state. Nevertheless, the non-unitary evolution in the deterministic part of the stochastic sampling conserves the total magnetization and only the jump applications switch between different symmetry blocks, so that the MCWF evolution can be calculated using conserved quantum numbers.

While the initial state was a product state in the previous chapter, the system is here initially prepared in the ground state of the equilibrium model accessed with the density ma-

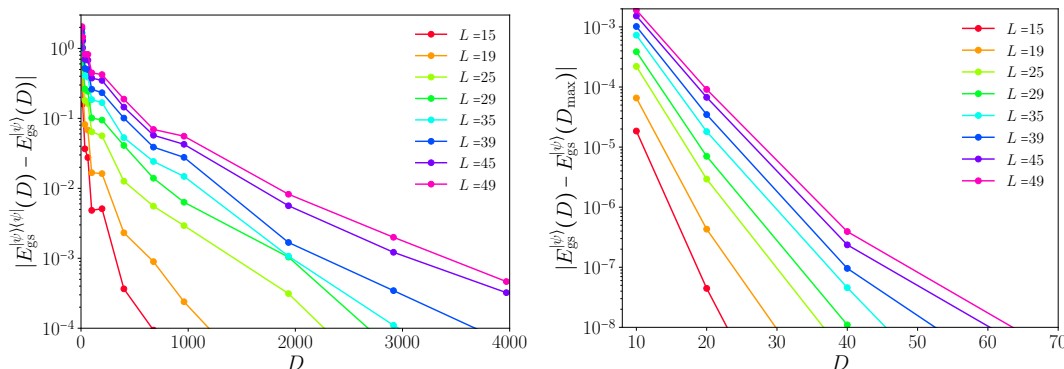

Figure 17: (left panel) Deviation of ground state energy after purification from the value obtained by DMRG in the original Hilbert space for different system size $L$. (right panel) The difference between ground state energy obtained by DMRG in the original Hilbert space for two bond dimensions $D$ and $D_{max}$. The maximum bond dimension fulfills truncation error $\epsilon = 10^{-12}$ and varies from $D_{max} = 40$ for system size $L = 15$ to $D_{max} = 90$ for system size $L = 49$. The ground state has been calculated for $J_z/J_x = 0.5$.

trix renormalization group (DMRG) ground state search algorithm [56, 72]. We have chosen the ground state of an anisotropy of $J_z/J_x = 0.5$, which is located in the gapless Tomonaga-Luttinger liquid phase of the Hamiltonian and requires a sizable bond dimension for its representation in MPS form. As a result, a strong truncation is necessary when reshaping the corresponding density matrix to a purified state in the doubled Hilbert space as described in Sec. 3.2.1. In Fig. 17 (left panel) we show the bond dimension dependence of the deviation of the ground state energy calculated after the reshaping process from the original value obtained by DMRG. For small system sizes with less than 20 sites, the ground state energy can be reproduced after the purification step with a medium sized bond dimension with less than $10^3$ states taken into account. However, even for slightly larger systems with up to 50 spins bond dimensions of several thousand states are needed to achieve an accuracy of the ground state energy of only $10^{-3}$. In this case the purification step alone takes more than three days of run time. On the other hand in Fig. 17 (right panel) the energy difference between two ground states in the original space with two different bond dimensions $D$ and $D_{max}$ is plotted for different system sizes. One sees that without purification the bond dimension below 100 is large enough to get the convergence of ground state energy of $10^{-8}$.

To estimate the numerical effort and the influence of inaccuracy in the representation of the ground state on time evolution for the purification method we compare the equal-time correlation functions for different values of the bond dimension in Fig. 18. As the equal-time correlations represent the initial condition for the two-time correlation functions at time $t_1$, the accuracy of their calculation is crucial in order to obtain the two-time correlation functions. The direct comparison in Fig. 18 shows that the purification method reaches a comparable accuracy to the MCWF approach for $D \geq 600$. Even for these bond dimensions sizable deviations occur of the order of $10^{-3}$. Looking at the associated run times, as summarized in Tab. 1, shows that the simulation based on purification takes about ten times as long for such a large bond dimension as compared to a parallel implementation of the stochastic sampling.

Based on the substantially larger run time (of a few weeks to months), the purification

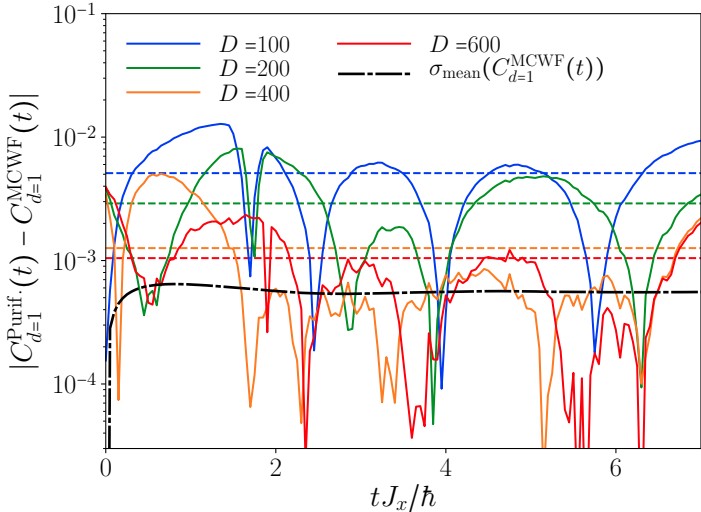

Figure 18: Comparison of the evolution of the equal-time correlation function under the dissipator $\mathcal{D}_2$. We present the deviation of the simulation results using the purification approach with different values for the maximal bond dimension from MCWF data using $3 \times 10^4$ samples and $D = 500$ (solid lines). We evolve a chain of size $L = 29$ with $\hbar\gamma/J_x = 2$ and $J_z/J_x = 0.5$. The time step is chosen as $\Delta t J_x/\hbar = 0.05$ in all cases. The dashed lines mark the time average over the presented time interval and the dashed-dotted line is the time-dependency of the standard deviation of the Monte-Carlo average.

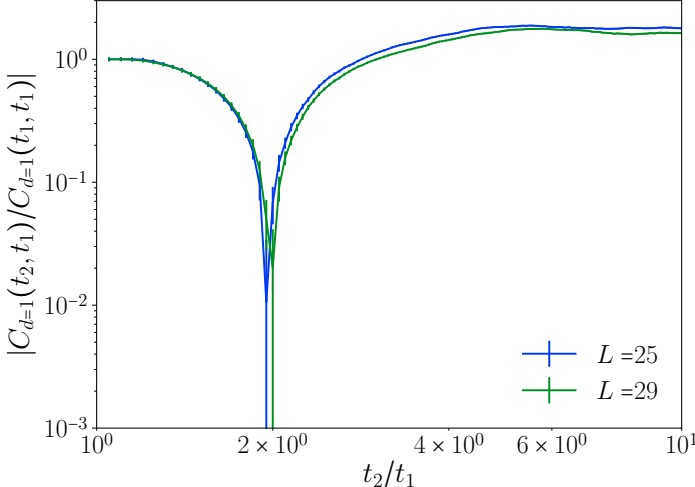

Figure 19: Evolution of the normalized two-time correlation functions computed by the MCWF method by Breuer et al. The simulation has been done for $\hbar\gamma/J_x = 1$, $J_x/J_x = 0.5$, $D = 500$, $t_1 J_x/\hbar = 1$, $\Delta t J_x/\hbar = 0.05$ and $10^4$ trajectory samples.

becomes very inefficient for the analysis of a physical situation. Even though the actual run time on a single core would be comparable, here, the MCWF scheme is preferential since the 'waiting time' to obtain the results makes a thorough analysis of a physical question

| method | bond dimension | run time [hours] |
|--------|---------------|------------------|
| MCWF | 500 | 84.47 |
| purification | 100 | 2.93 |
| purification | 200 | 29.04 |
| purification | 300 | 110.28 |
| purification | 400 | 355.24 |
| purification | 500 | 818.38 |
| purification | 600 | 829.88 |

Table 1: Table of run times for evaluating equal-time correlation functions from Fig. 18. The MCWF simulation was executed in parallel on 10 cores. Parameters are the same as in Fig. 18.

more feasible. The waiting time to obtain the results can be easily shortened by the trivial parallelization of the stochastic approach.

To conclude the analysis, it remains to be demonstrated that the two-time correlation functions are accessible by the MCWF scheme. For this purpose we show in Fig. 19 the two-time correlation functions for the system sizes $L = 25$ and $L = 29$ which would be very inaccurate and time consuming to reach by the purification method. We see that the two time function is rising in time, which signals rising fluctuations. A detailed study of the arising physics goes beyond the present work with a strong technical focus.

# 6   Conclusion

To conclude we have presented a comparison of three different MPS based methods for the calculation of two-time functions in open quantum systems. This comprises the purification approach and two different approaches based on the stochastic unraveling of the Lindblad dynamics. First we compared the two stochastic approaches in the situation of an XXZ spin chain subjected to a dephasing noise starting initially in the classical Neel state. In this situation we find a clear preference for the stochastic approach suggested by Breuer et al. over the approach suggested by Mømler et al.. This is due to the better convergence of the trajectories used in the approach by Breuer et al.. However, the purification approach is even much more efficient for the considered situation. We would like to emphasize that this conclusion that the purification approach was the most efficient also hold for correlations of the type $S^+S^-$. Additionally, we considered the dynamics of a XXZ spin chain subjected to a local application of the jump operator $S_c^-$ starting from a Tomonaga-Luttinger liquid. This changes drastically the efficiency of the methods and the stochastic approach becomes more efficient than the purification approach. There are several reasons for this. First, already the representation of the Tomonaga-Luttinger liquid in the purified form is resource demanding. Secondly, in the following time-evolution the conservation of the magnetization is not fulfilled anymore. These reasonings also hold for correlations of different type. We therefore expect that the purification approach is valuable if the initial state is easily represented within the purified space. Further, a strong symmetry of the Linbladian enabling the use of conserved quantities is of advantage. In comparison the stochastic wave function approach is well suited also to represent difficult initial states and the following Lindblad evolution. However, the presence of many jumps, as for the case of strong dephasing noise, calls for a very low time-

step which makes the trajectory approach less efficient. We would like to point out that even though the comparison was mainly performed for short range correlations $d = 1$, since the computational efficiency up to the application of the operator at time $t_2$ is independent of the chosen distances, all our findings will also hold for larger values of the distances.

# Acknowledgements

We thank J. Dalibard, M. Fleischhauer, A. Läuchli, and K. Mølmer, for fruitful discussions.

**Funding information**    We acknowledge funding from the Deutsche Forschungsgemeinschaft (DFG, German Research Foundation) in particular under project number 277625399 - TRR 185 (B4) and project number 277146847 - CRC 1238 (C05) and under Germany's Excellence Strategy – Cluster of Excellence Matter and Light for Quantum Computing (ML4Q) EXC 2004/1 – 390534769 and the European Research Council (ERC) under the Horizon 2020 research and innovation program, grant agreement No. 648166 (Phonton).

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
