# Peer review of "Numerical evaluation of two-time correlation functions in open quantum systems with matrix product state methods: a comparison"

_SciPost Physics Core_

## Round 1 · Referee Report · Anonymous · 2020-5-7

Strengths

1. Comparison of three different approaches to calculate the two-time correlation function for open systems are discussed.
2. Two different forms of dissipation are investigated.
3. A self containing introduction into the used approaches is given.
4. A method to purify a given MPS is given.

Weaknesses

1. Not all approaches are investigated for both forms of dissipation.
2. Notation is inconsistent and several objects are not explicitly defined.
3. Figures and captions are sometimes not fully syncronized.

Report

In "Numerical evaluation of two-time correlation functions in open quantum systems with matrix product state methods: a comparison" the authors compare three approaches (one based on the purification approach and two based on Monte-Carlo wave functions (MCWF)) to calculate two-time correlation functions in open quantum systems. Some approaches are applied to two different dissipation setups.

The manuscript delivers a good, self contained, in-depth overview of the approaches in question and the goal of having a comprehensive comparison is certainly interesting for everyone who wants to analyze open quantum systems with MPS.

Nevertheless, there are several open questions, some recommendations, and some requested changes that need to be answered prior to a possible publication. The question and the recommendation are listed directly below and the requested changes are listed in the separated part. Every list is ordered by the appearance within the manuscript, not by relevance.

Questions:

1. The upper half of page 10 seems to be already discussed in Ref. 47. While it is certainly good practice to repeat important technical details, it is maybe helpful to stress - in this case - that it is done in order to explain the transformation of the Lindbladian.

2. In Fig. 18 the authors show the deviation of several purification approach calculations from the MCWF data and indicate the standard deviation of the Monte-Carlo average as a guidance for the error of the MCWF. In Fig. 13 the authors showed that the statistical error is much smaller than the truncation error for the bond dimension used in Fig. 18. Despite the fact that different correlation functions (equal-time vs. two-time) were measured, it is not clear why the truncation error seems to be of no interest in this comparison.

3. A closer look at Table 1 shows that the purification approach mainly takes longer because of the lack of symmetries and/or the (principled) lack of parallelism. If compared "fairer" (both approaches with a single core) the purification might even be faster. Especially because it might even be converged with a bond dimension of 600. Is that assessment correct and if so, why was this representation chosen instead?

Recommendations:

1. It would be helpful to explain why it is fairer to compare a special case instead of the more general one.

2. As the approach by Mølmer et. al. builds up entanglement faster than the approach by Breuer et. al. it might be interesting to see how both approaches handle non exact MPS.

3. It would be interesting how the approach by Mølmer et al. compares to the approach by Breuer et al. in the case of the second dissipation setup. The authors assumably have done such calculations and it would be good to see something like "The approach by Breuer et al. shows in this setup similar behavior as in the previous setup...", if that turns out to be the case. Otherwise a more detailed explanation for not including this approach in the comparison might be necessary.

Requested changes

1. Please carefully check that references actually contain the claimed content. In particular reference 1 is referenced multiple times, but the promised informations seem not to be within the referenced paper (e.g., purification approach; two-time correlation functions (for spin-1/2 chain with dissipator $D_1$); a prove that two-time correlation functions are essential to uncover interesting dynamical regimes that display physical phenomena such as aging or hierarchical dynamics; the required time step to obtain good accuracy.).

2. Specify $t_1$ in the two-time correlation functions and if somehow possible include (maybe as additional material) some absolute data in order to establish reproducibility.

3. Check all lists of indices and alike whether or not commas are necessary/wanted. In the last line of Eq. 14 a mixture is used. Otherwise explain the chosen notation.

4. The sentence right above Eq. 5 seems to be incomplete.

5. Consider adding "Time-evolution methods for matrix-product states" (https://doi.org/10.1016/j.aop.2019.167998) to the list of citations concerning time-dependent matrix product states [47-50].

6. The $\hbar$ on the left hand site are missing in Eq. 12. Also the system size is usually not a part of the Trotter error, it should be removed or its origin and importance should be explained.

7. Is Fig. 2/Eq. 14 a figure or an equation? It should probably not end with a full stop.

8. Consider adding some original thermofield papers (pre MPS) to the list of citations concerning the doubled Hilbert space [28,47,53].

9. Please define the new $L$ and $D$ in Eq. 16 explicitly.

10. Please explain the choice for the black dashed and dotted lines in the caption of Fig. 4. What did you use as maximal accepted probability in the following examples?

11. The error bars in Fig. 5 are not resolvable and hence not visibly time dependent. Please consider either enhancing the scale of the error bars or removing them and state that the statistical errors are within the line width.

12. In the last sentence of point two on page 15 it should state "Here the norm of the new state is the same as the initial state".

13. In the last equation of point 4 on page 15 a ")" is missing on the right hand site.

14. The dashed lines in Fig. 9 are barely recognizable as dashed. Please use a different type of dashing or another way to seperate the maximum values from the averages. And mention them and their relevance in the caption.

15. Please elaborate what exactly can be learned from Fig. 19. As there is (apparently) no way for a comparison, can the results be discussed qualitatively?

16. Check reference 25 "[...] M. u. u. u. u. Guţă [...]"

  • validity: high
  • significance: good
  • originality: ok
  • clarity: good
  • formatting: reasonable
  • grammar: good

Author:  Ameneh Sheikhan  on 2020-09-16  [id 974]

(in reply to Report 1 on 2020-05-07)

$---------------$ our reply to Questions: $---------------$

1) We added a sentence to the text which makes clear that the chapters on MPS in closed systems and open systems is a review of the technique needed in order to understand the details of our implementation.

'In this section and the following on open systems, we describe the basics and key concepts of this technique [56] such that in the following we can detail the particularities of the approach to the determination of two-time functions in open quantum systems.'

2) We thank the referee for pointing out this confusing representation which we tried to clarify in the new version. In Fig. 13 we show that for a large bond dimension (approximately above D~ 400) the stochastic error becomes of the same order as the error measured by $\Delta$. This is due to the fact that $\Delta$ not only results from the truncation error, but also measures the stochastic error. This is due to the fact that we cannot generate as a comparison trajectory the 'same' probabilities. Therefore, the error $\Delta$ always has some of the stochastic error in it (but not in the same way as $\sigma$) and becomes for small truncations or large bond dimensions of the same order as $\sigma$. Thus, for large bond dimension the stochastic error is a good measure of the uncertainty of the curves. In Fig. 18 the standard deviation of the MCWF gives a good estimate of the size of the error.

We added the following explanations to the text:

'Let us note that this measure for the error not only measures the 'pure' truncation error, since the trajectories with a different bond dimension also have a different stochastic nature.'

'As can be seen in Fig. 13, increasing the number of samples used, the sampling error decreases. Further, with increasing bond dimensions, the truncation error becomes less important and $\Delta_{\mathrm{max}}(D)$ becomes of the same order as $\sigma_{\mathrm{max}}(D)$.'

3) We agree with the referee that how to present and compare the run time in a 'fair' way was not obvious. We had very long discussions on how to present these results in the manuscript. We have chosen this representation due to the long run times of the purification which make an efficient working with the data more difficult. The purification runs which are of sufficient accuracy take between several weeks up to more than a month in order to reach the time $t_1$. After this, the calculation of the correlation has to be started which then adds another sizable duration to the actual run time. Therefore, to obtain the results for one parameter set takes up to several months and to reach longer system sizes would be very lengthy. A month is a long time on the work time of a Master or PhD thesis. The possibility to trivially parallelize the MCWF runs for the different trajectories (typically much more than 100 cores can be used) makes it much more efficient in order to really investigate a physical problem.

This difficulty to make the comparison 'fair' is the reason why we have chosen to also give the actual run times even though this typically strongly depends on the actual computational resources and the exact implementation. However, in the present situation we expect that the reader can use these run times in order to judge him/herself the numerical effort and which method would be appropriate for his/her purpose.

We have reformulated the comparison:

'Based on the substantially larger run time (of a few weeks to months), the purification becomes very inefficient for the analysis of a physical situation. Even though the actual run time on a single core would be comparable, here, the MCWF scheme is preferential since the 'waiting time' to obtain the results makes a thorough analysis of a physical question more feasible. The waiting time to obtain the results can be easily shortened by the trivial parallelization of the stochastic approach.'

$---------------$ our reply to Recommendations: $---------------$

1) In the approach by Breuer et. al. there are two trajectories evolving in time where the quantum jumps occur jointly. On the other hand in the approach by Mølmer et. al. in general there are four trajectories evolving in time independently. Just counting the number of trajectories, this would naively lead to the conclusion that the approach by Mølmer et. al. is of double numerical cost. In the considered special case (A being hermitian and [A,B]=0), two trajectories (with independent jumps) are enough for the approach by Mølmer et al. which is the same number as in the Breuer et. al. approach. Since even in this very favourable situation for the Mølmer et. al approach we find that it is numerically more costly than the Breuer approach, it will be even more costly in the more general situation where 4 trajectories are necessary.

We added a sentence in order to explain this 'To ensure that the comparison between the two approaches is as fair as possible, we concentrate on such a special case for the two-time correlator in Eq. 45, since for this case the number of trajectories needed in both approaches is equal. We expect that in the more general case where even four trajectories are needed in the approach by M\o lmer et al., this approach will become even more costly.'

2) We have added two panels considering larger system sizes, where the truncation of the MPS has been performed. We added the following paragraph.

'In the right panel of Fig. 9 larger system size $L=20$ are shown. The total von Neumann entropy generated ($J_z/J_x=2$) is larger for larger system sizes. However, concerning the behaviour of the different approaches the findings of the small system size are confirmed that the von Neumann entropy in the M\o lmer approach grows a bit faster than the one of the approach by Breuer et al..'

3) We have only compared the approaches by Mølmer et al. to the approach by Breuer et al. in the case of the first dissipation. The reason for this is that the second dissipation setup would require that for the Mølmer et al. approach we evolve 4 different trajectories, since the simplification explained for the D1 does not hold any more. Since from previous findings, we do not expect the computational complexity of each of these trajectories to be only half of the two trajectories required for the Breuer approach, we did not perform a comparison.

$---------------$ our reply to Requested changes $---------------$

1) We thank the referee for pointing this out. The label to the reference was the same as for another reference. Therefore, it was referenced wrongly in the manuscript. The correct reference 1 is the paper by the authors entitled "Evolution of two-time correlations in dissipative quantum spin systems: Aging and hierarchical dynamics" cites as Phys. Rev. B 100, 165144 (2019). We corrected this in the revised article.

2) We provide the values of $t_1$ for each of the plots to ensure reproducibility.

3) We thank the referee for pointing out that we had inconsistencies. We removed the commas in the notation.

4) we completed the sentence: 'The dissipation strength is $\Gamma_{c}=\gamma$ and $\Gamma_{l\neq c}=0$ which results in the dissipator' ...

5) We added the reference.

6) We thank the referee for pointing out that typo which we corrected.

7) We have removed the label Eq. (14) and left it as a Figure.

8) We added new references: [27] T. Nishino, Density Matrix Renormalization Group Method for 2D Classical Models, J. Phys. Soc. Jpn. 64, 3598 (1995), doi:https://doi.org/10.1143/JPSJ.64.3598. [28] S. Moukouri and L. G. Caron, Thermodynamic Density Matrix Renormalization Group Study of the Magnetic Susceptibility of Half-Integer Quantum Spin Chains, Phys. Rev. Lett. 77, 4640 (1996), doi:= 10.1103/PhysRevLett.77.4640. [29] D. Batista, K. Hallberg, and A. A. Aligia, Specific heat of defects in the Haldane system Y2BaNiO5, Phys. Rev. B 58, 9248 (1998), doi:10.1103/PhysRevB.58.9248. [30] K. Hallberg, D. Batista, and A. A. Aligia, Specific heat of defects in the S = 1 chain system Y2 BaNiO5 , Physica B: Condensed Matter 261, 1017 (1999), doi:= https://doi.org/10.1016/S0921-4526(98)00614-0.

9) We substituted the '$=$' signs by the '$\equiv$' signs in order to make the definition more explicit.

10) We added a sentence to the caption of Fig. 4 explaining the black lines which represent the typically chosen time steps for the given examples.

'The vertical dashed lines mark the time-steps typically chosen in the calculations and the horizontal dotted lines are the corresponding probability for $L=16$.'

11) The error bars are too small to be visible. We added a remark to the caption.

12) We corrected the sentence as 'Here the norm of the new state is the same as of the initial state ($\langle{\tilde{\Theta}(t_1)} |\tilde{\Theta}(t_1)\rangle = \langle{\psi(t_1)}| \psi(t_1)\rangle$).'

13) We corrected this misprint.

14) The plot is updated with larger dashed lines.

15) From the methodological side, we prove with Fig. 19 that it is currently possible to perform two-time calculations for the considered situation using the MCWF approach for reasonable system sizes. We added a sentence on the physics which can be extracted from these results. However, since many more runs at different parameters would be required for a more thorough study of the physics, this is left for another work.

'We see that the two time function is rising in time, which signals rising fluctuations. A detailed study of the arising physics goes beyond the present work with a strong technical focus.'

16) We corrected the reference.

---

## Round 1 · Referee Report · Matteo Rizzi · 2020-5-14

Strengths

1- The manuscript presents a complete, self-contained guide about three different methods to tackle an highly-relevant task, namely the study of open quantum systems dynamics.

2- The three methods are carefully compared in terms of accuracy and computational performances, in order to convey the message that the method of choice strongly depends on the model investigated. Guiding criteria to generalise the choice beyond the specific example are provided.

3- The written text is nicely complemented by informative plots and tables of high readability.

Weaknesses

1- Some important references appear to be missing here and there (see Report): nothing too fatal but it should be amended.

2- There are some technical aspects that to be clarified, mainly about the exploitation of symmetries and the performed benchmarks: e.g., changing two relevant aspects at the same time, Lindblad operator and initial state, leaves some questions open to my eyes.

3- The large distance between the model description (Sec.2, pages 3-4) and the discussion of the method comparison (Sec.5, pages 17 -on) seems suboptimal to me. It is probably matter of taste, but I would have found more reasonable to have the general descriptions of the methods (Sec. 3-4) earlier, since they are actually almost independent of the chosen model.
Some parts could have as well be moved into appendices.

Report

I have no doubt about the overall quality of the performed studies and the aim of the manuscript, as visible from the list of “Strengths” above. The manuscript could be publishable almost in its present form, except for a couple of extra references to be integrated into it.
Nonetheless, I would invite the Authors to consider the “Remarks” listed here below (and hinted to in the “Weaknesses”) as primarily aimed at further improving the impact of the manuscript as a (maybe "the") reference guide for people working in this relevant field.
* * *
References
* * *
(important) Alongside with Ref. [30], I would have expected to see the equally founding R. Dum, P. Zoller, and H. Ritsch, Phys. Rev. A 45, 4879 (1992). By the way, why is [30] not cited together with [3]?

(others) Here some references about tensor-network methods for open quantum systems, which I was quite surprised not to find cited in the overview section of the present manuscript — they are not necessarily the only ones, and should be intended as a suggestion for a more exhaustive collection of references:
T. Prosen, and M. Znidaric, Phys. Rev. E 75 015202(R) (2007);
M.J. Hartmann, et al. Phys. Rev. Lett. 102 057202 (2009);
A.H. Werner, et al., Phys. Rev. Lett. 116, 237201 (2016).

(minor) In the context of MonteCarlo-MPS methods, I would have said that Ref. 57 by A. Daley would have deserved a more prominent role, not? Maybe also some specific papers by him and co-authors (around 2010 in Zoller’s group), though a direct citation to the latters might probably be circumvented by a “… and references therein” in Ref. 57.
* * *
Technical Remarks
* * *
a) About methods for computing efficiently the time-evolution of purified states (Sec. 4.1), I wonder whether the methods introduced for thermal ensembles in the following works might be of any help here for Lindbladian dynamics, too:
C. Karrasch, J. H. Bardarson, and J. E. Moore, Phys. Rev. Lett. 108, 227206 (2012) & New J Phys 15, 083031 (2013);
T. Barthel, New J. Phys. 15 073010 (2013) & related works;
I. Pizorn, et al., New J Phys 16, 073007 (2014) & related works.

b) About purification in presence of conserved quantities (Sec. 3.2.4), I wonder whether the following works might be of relevance for the Authors’ sakes here (though they also mainly deal with thermal canonical ensembles only, not open dynamics with jumps…):
A. Nocera, G. Alvarez, Phys. Rev. B 93, 045137 (2016)
T. Barthel, Phys. Rev. B 94, 115157 (2016)

c) Incidentally, why do the Authors not introduce directly the convenient trick of Eq.(25) already at the level of representing the initial state (Sec. 3.2.2), since they claim that it brings advantages also in that task? This would help to stream-line the reading for the non-insider reader, similarly to what is nicely achieved in almost the whole rest of the manuscript.

d) Moreover, does the same trick not allow for the conservation of the total magnetization in the transformed purified state, $| \rho \rangle\rangle$? I mean that the new $\mathbb{D}_2^{\prime} \equiv \mathbb{U}^\dagger \mathbb{D}_2 \mathbb{U}$ should then read $\mathbb{D}_2^{\prime} = \gamma \left(S_c^- \otimes S_c^+ - \frac{1}{2} \left(\frac{\mathbb{I}}{2} + S^z\right) \otimes \mathbb{I} - \frac{1}{2} \mathbb{I} \otimes \left(\frac{\mathbb{I}}{2} - S^z\right) \right)$, if I am not terribly mistaken, in which case I apologize. Does it not bring at least some computational advantage, though not as much as a separate conservation of $S^z$ in both bra and ket spaces? If this has been already exploited, could the Authors convey the information in a more accessible way?

e) The benchmark for $\mathcal{D}_1$ are performed starting from a classical product state, while the ones for $\mathcal{D}_2$ from the actual, very entangled, quantum ground-state in a gapless regime.
I would be curious to know what the performances of the two (three) approaches would be for $D_1$ starting from the latter initial state. Is there a line of reasoning to spare the actual calculation and already draw conclusions from the present data?
Sorry in advance, if the questions sounds too naive to insiders.
Incidentally, among the MonteCarlo variants, is the Breuer et al. method guessed to be always better than the Mølmer et al. method, in whatever situation? Or can one at least speculate a situation (no need to prove it here and now) in which the latter would be more convenient?

f) The comparison is limited to two-point observables of a very special kind, namely $\langle S^z_{c}(t_1) S^z_{c+d}(t_2) \rangle$, where both operators are Hermitian and commute with each other. Equally interesting would be to assess performances for the computation of $\langle S^+_{c}(t_1) S^-_{c+d}(t_2) \rangle$ (and maybe their fermionic counterpart with Jordan-Wigner string in the middle).
It is fully clear to me that this would cause the loss of some important simplifications here and there, and probably result into a more expensive set of runs: nonetheless, I really think that this would considerably enlarge the content, and therefore the relevance, of this manuscript for people working (or wishing to enter) in this relevant research field.
If a clear statement on these observables could be already provided based on present data only, instead, this should be given a prominent role in the discussion section (without need for burning further computational time).
* * *
Other minor comments
* * *
A) Punctuation should be double checked again.
E.g., at page 4, "This section is structured such, that we start by giving a ..." would need no comma, in my humble opinion. Several similar examples are disseminated in the text.

B) In describing the properties of $\mathcal{D}_1$, it is stated that the jump operators being Hermitian implies that the infinite temperature state is the unique steady state of the model.
Am I right in saying that, more generally, one simply needs that the jump operators are normal (i.e., $[L,L^\dagger]=0$) to ensure the $T=\infty$ state to be a steady state, $D(\mathbb{I}) = 0$?
What about the unicity? Can the Authors sketch a proof for it?

C) While describing the general formalism of MPS and their (in principle, exact) construction via iterative SVDs, would the resulting Eq. (7) not be a MPS in the canonical $(\Gamma, \lambda)$ form? Since the aim of the explanation is clearly a didactical one for newcomers, I would not skip logical steps :-)

D) The placement of some figures (e.g., 2-3, but also others) is a bit unfortunate with respect to where they are referenced to in the main text. If possible, this should be amended.

E) Should the compression of the purified state (around Eq. 18) be performed while sweeping and keeping the canonical form with the target site as the orthogonal centre of the gauge? A brief comment would be of help for the reader.

F) Actually, another family of approaches exist on the market, namely the one based on matrix-product density-operators (MPDO), e.g., Ref. 27 & 56 (but not only): a comment about the advantage/disadvantage of the purification approach with respect to a direct MPDO approach would be nice to have here.

G) In Fig. 5, did something went wrong with the formatting? It is difficult to believe that the statistical error bar is not visible on this scale, not? Or am I dumbly mislead?

H) All comparisons are performed for extremely short distances in the correlation functions (d=0 or 1): it could be interesting to know, at least speculatively, whether and how larger d's would affect the statements drawn from the benchmarks.

I) In Fig.. 13, why are some data bouncing in a non-monotonic direction at large bond-dimensions? Simply lack of statistics (larger samples $R$ seem to be less affected), or is there something more subtle?

L) In the error estimate of Eq.(49) I would have naively thought to some quadrature sum, instead of a max(...). This would certainly not change the comparison outcome for $\mathcal{D}_1$, but what about $\mathcal{D}_2$ (let's forget for a moment about run-time)?

M) In Fig. 17, it could be useful for newcomers to plot the same error estimate for the pure-state representation alone (errors should be considerably smaller for the given $L$ and $D$'s): this would give immediate reason for the chosen D=500 for the MCWF method as a reference...

N) In Table 1 it is mentioned that "the MCWF simulation was executed in parallel on 10 cores": does it mean that the actual CPU-time would be 10 times larger, making this comparable to the purification one (845 vs 830 hours, if so)? I.e., please indicate whether the quoted run-time is CPU or wall-clock :-)

O) In Sec. 4.2.2 it is strange to read "An alternative strategy, established by Mølmer et al. [3], uses the quantum regression theorem [31]", since it sounds like an article of 1992 using a theorem of 2003...

  • validity: high
  • significance: good
  • originality: good
  • clarity: high
  • formatting: good
  • grammar: good

Author:  Ameneh Sheikhan  on 2020-09-16  [id 975]

(in reply to Report 2 by Matteo Rizzi on 2020-05-14)

$---------------$
our reply to comments on References
$---------------$

(important) We include the reference suggested by the referee. In the abstract we preferred to refer only to Ref.[3] which explains the method and in particular the calculation of the two-time functions in detail.

(others) We added 2nd and 3rd references suggested. We cannot provide here a full list of articles which have used tensor-network methods for open quantum systems. We have restricted ourselves to the ones which had methodological advances used in this work.

(minor) We thought we do cite it prominently at the place where we explain that the stochastic wave functions approach can be combined with the MPS time evolution. We have changed this to '(see [70] and references therein)' and we have now additionally added the paper to the citations in the introduction.

'For the determination of a single trajectory applying the introduced $t$MPS algorithm Eq. 13 offers a promising solution for computing the deterministic part of the evolution (see [70] and references therein).'

'and two different stochastic approaches based on the unraveling of the quantum evolution [2,3,36-39,70].'

$---------------$
our reply to Technical Remarks
$---------------$

a) We thank the referee for pointing out these papers which we are certainly acquainted with. However, the freedom of the transformation present for the thermal evolution is not present for the Lindblad evolution. Therefore, one cannot transfer the approach straightfowardly to the Lindblad evolution. In order not to confuse the readers, we have not added the references to the evolution of the thermal states.

b) We thank the referee for pointing out these works. However, we decided not to add them, since they are focused on the thermal ensembles.

c) We agree with the referee that in principle one could also introduce it in section 3.2.2. However, we decided to present the transformation (Eq. 25 and following) only after introducing the calculation of the expectation values. Our reasoning for this is that (i) the unitary transformation is not needed for a situation in which no good quantum numbers occur and (ii) that only the problem to represent the trace (the 'one') in an efficient way explains why the transformation is so useful. We therefore prefer to leave the explanation in the section of the calculation of the expectation values.

d) In the case with dissipator $\mathcal{D}_2$ we do not have the conservation of total magnetisation in the purified state (neither in the unitary transformed purified state). In particular, if we start with the totally polarized up state, the final steady state is still the totally polarized down state.
If we understand the reasoning of the referee correctly, we think that maybe there is a mistake in applying the transformation. We obtain the expression:
$\mathbb {D}′_2=\gamma\left(S^−_c\otimes S^-_c −\frac{1}{2}(\frac{I}{2}-S^z_c)\otimes I−\frac{1}{2}I\otimes(\frac{I}{2}−S^z_c)\right)$
for the transformed purified state which would not conserve the total magnetisation.

e) Starting $\mathcal{D}_1$ from the quantum ground state approach with purification is doable but still very demanding. The computational effort for the purification compared to case $\mathcal{D}_2$ is less, since here good quantum numbers simplify the representation considerably. However, already the accurate representation of the ground state is despite the presence of good quantum numbers highly challenging. Such simulations were performed for the equal time correlations in arXiv:2003.13809. However, we expect that the stochastic calculations would be much easier in this situation for most parameters sets in order to extract the interesting physics.

f) We focused our comparison to the $S^zS^z$ correlation function since this is the correlation which shows the most interesting dynamics caused by the competition of the closed system dynamics and the dissipation. The correlations $S^+S^-$ are dominated by the dissipative dynamics and show an exponential decay with time-scale gamma as discussed in https://arxiv.org/pdf/1809.10464.pdf and results shown in the PhD Thesis of S. Wolff (http://hss.ulb.uni-bonn.de/2019/5459/5459.pdf).

In terms of computation the correlation $S^+S^-$ again the conservation of the magnetization can be applied both in the purification approach and also in the stochastic approach. The computational effort was found to be less or similar than the $S^zS^z$ correlations for the purification approach. The stochastic approach became a bit more intricate, since due to the very fast decay of the correlations the stochastic variations made a good determination of the very small value difficult. Therefore, the conclusion drawn from our studies were very similar to the ones for the $S^zS^z$ correlations. This was the reason for us to focus our study on this most interesting and most delicate correlations $S^zS^z$.

For the $\mathcal{D}_2$ dissipation, the problem of the purification approach is mainly attributed to the purification of the initial state. Here also the same complexity is expected for the correlation $S^+S^-$. For the stochastic approach again, we do not expect that this correlation will lead to a very different behavior in the computational complexity.

We added a discussion to the conclusion:
'However, the purification approach is even much more efficient for the considered situation. We would like to emphasize that this conclusion that the purification approach was the most efficient also hold for correlations of the type $S^+S^-$.'
'First, already the representation of the Tomonaga-Luttinger liquid in the purified form is resource demanding. Secondly, in the following time-evolution the conservation of the magnetization is not fulfilled anymore. These reasonings also hold for correlations of different type.'

$---------------$
our reply to Other minor comments
$---------------$

A) We corrected the sentence. We tried to correct similar sentences. However, non of us has English as its mother tongue which complicates this task for us.

B) To obtain that the infinite temperature state is a steady state, one needs that $LL^\dagger- L^\dagger L$ vanish. As the referee correctly states, it is fulfilled if the jump operators are normal. The uniqueness of the steady state is more complicated and had been discussed in the Refs.[34–36]. The reasoning is that all other steady states in the dissipation free subspace are connected by the hopping event to higher dissipation subspaces such that these states decay (using many body adiabatic elimination). Therefore, only the infinite temperature state is the steady state. These arguments are further supported by numerical findings and calculations for the non-interacting system.

C) We do not aim at a fully self-explanatory review of the basics of the MPS methods, since many very nice reviews are available for this task and we do not want to repeat this here. More importantly we do aim to provide a self-contained introduction to the dissipative methods and their limitations. In particular, newcomers to the dissipative methods should be able to follow easily our descriptions. With the section of the general formalism of the MPS we mainly want to explain the approximations made within the method and to set the language which we use in the following. We hope that this explains our choice of the description. If the referee feels that another formula would needed for this, we are happy to further consider this if he could specify.

D) we moved the figures to be closer to the corresponding text.

E) As explained in the paragraph after eq. 20, we performed the compression already in the states which need to be purified. This choice was done in order to simplify the computational effort.

F) We have not implemented such an approach for the calculation of two time functions. Therefore, we cannot make any firm statements on this approach for two-time functions. However, from our understanding also in the thermal case, the 'normal' MPS approach can be advantageous over the MPDO approach, such that we expect this also for the dissipative systems which justifies our current study.

G) The standard deviation (see Eq. 32) is small because of the very large number of samples ($R=10^4$) chosen.
In order to clarify this, we added the comment '(not visible, since they are below the line width)' to the caption of Fig. 5.

H) The dependence on the distance of the correlations only enters in the last step of the calculation, the evaluation of the expectation values. Therefore, the computationally most expensive steps, the intermediate time evolution after the application of the operator at time t1, does not depend on the chosen distance. Therefore, all the findings hold up to the application of the operator at time t2 and only the computational cost of the evaluation of following expectation value rises slightly due to the longer distance.
We added a sentence to the conclusion.

'We would like to point out that even though the comparison was mainly performed for short range correlations $d=1$, since the computational efficiency up to the application of the operator at time $t_2$ is independent of the chosen distances, all our findings will also hold for larger values of the distances.'

I) The data bouncing happens when the error $\Delta_\mathrm{max}$ becomes of the same order of the stochastic error sigma. As the referee mentions, we attribute this to the stochastic effects which are within the error $\Delta_\mathrm{max}$ due to the stochastic nature of both averages taken at different $\Delta_\mathrm{max}$ . Since here a difference is taken, the statistics might become more important.

L) We have chosen here the maximum, since we are typically in a situation where one of the errors is dominating. We agree with the referee that the quadrature sum could be another choice. However, we do not expect that any of the conclusions drawn from the error comparison would change. As the referee states the comparison for $\mathcal{D}_1$ would not change much. Since for $\mathcal{D}_2$ we do not use this definition for the comparisons we do not fully understand what the referee aims on with the comment.

M) We have added a plot showing the same error estimate for the pure-state representation and added a paragraph in order to comment this.

'On the other hand in Fig. 17 (right panel) the energy difference bet ween two ground states in the original space with two different bond dimensions $D$ and $D_{\text{max}}$ is plotted for different system sizes. One sees that without purification the bond dimension below $\sim 100$ is large enough to get the convergence of ground state energy of $10^{-8}$.'

N) Yes, the actual CPU-time is 10 times larger and therefore comparable with the time of the purification approach. However, the 'waiting time' for the user which is important in order to have an efficient working on a project can be easily reduced by the availability of a suitable cluster for the MCWF method. We added a sentence

'Based on the substantially larger run time (of a few weeks to months), the purification becomes very inefficient for the analysis of a physical situation. Even though the actual run time on a single core would be comparable, here, the MCWF scheme is preferential since the 'waiting time' to obtain the results makes a thorough analysis of a physical question more feasible. The waiting time to obtain the results can be easily shortened by the trivial parallelization of the stochastic approach.'

O) We added the reference to the original paper on the quantum regression theorem:
[71] M. Lax, Formal Theory of Quantum Fluctuations from a Driven State, Phys. Rev. Lett. 129, 2342 (1963)
which clarifies the formulation.

---

## Editorial Decision

resubmitted